# Chloroplast and whole-genome sequencing shed light on the evolutionary history and phenotypic diversification of peanuts

Zheng Zheng [1,2,3,4,9] ✉, Ziqi Sun[1,2,3,4,9], Feiyan Qi[1,2,3,4,9], Yuanjin Fang[1,2,3,9], Ke Lin[1,2,3,9], Stefano Pavan[3,5,9], Bingyan Huang[1,2,3], Wenzhao Dong[1,2,3], Pei Du[1,2,3,4], Mengdi Tian[1,2,3], Lei Shi[1,2,3,4], Jing Xu[1,2,3], Suoyi Han[1,2,3], Hua Liu[1,2,3], Li Qin[1,2,3], Zhongxin Zhang[1,2,3], Xiaodong Dai[1,2,3], Lijuan Miao[1,2,3], Ruifang Zhao[1,2,3], Juan Wang[1,2,3], Yanlin Liao[3,4,6], Alun Li [7], Jue Ruan [7], Chiara Delvento [5], Riccardo Aiese Cigliano[8], Chris Maliepaard[6], Yuling Bai[6], Richard G. F. Visser [6] & Xinyou Zhang [1,2,3,4] ✉

Cultivated peanut (*Arachis hypogaea* L.) is a widely grown oilseed crop worldwide; however, the events leading to its origin and diversification are not fully understood. Here by combining chloroplast and whole-genome sequence data from a large germplasm collection, we show that the two subspecies of *A. hypogaea* (*hypogaea* and *fastigiata*) likely arose from distinct allopolyploidization and domestication events. Peanut genetic clusters were then differentiated in relation to dissemination routes and breeding efforts. A combination of linkage mapping and genome-wide association studies allowed us to characterize genes and genomic regions related to main peanut morpho-agronomic traits, namely flowering pattern, inner tegument color, growth habit, pod/seed weight and oil content. Together, our findings shed light on the evolutionary history and phenotypic diversification of peanuts and might be of broad interest to plant breeders.

Cultivated peanut or groundnut (*Arachis hypogaea* L.) is a sustainable and affordable source of edible oil and proteins, which globally yields 54 million tons from a cultivated area of 32 million ha (http://www.fao.org/faostat, 2020). Its allotetraploid nature (genome AABB, size ~2.7 Gb) is thought to arise from the polyploidization of an interspecific hybrid between 2 of 81 wild species, currently described in the genus *Arachis*—*Arachis duranensis* Krapov. and W.C. Gregory (genome AA, size ~1.25 Gb, female parent) and *Arachis ipaënsis* Krapov. and W.C. Gregory (genome BB, size ~1.56 Gb, male parent)[1,2].

*A. hypogaea* is commonly assumed to be domesticated from the wild tetraploid progenitor *Arachis monticola*, most probably in a region now encompassing part of southern Bolivia and northern Argentina[1,3–5]. The first archeological evidence of peanut cultivation traces back to 7,600 years ago[6]. In the 16th century, peanut cultivation diffused from South America through the Portuguese and the Spanish explorers[7]. Further migration routes from North America to Northern China and from South Asia to Southern China have been recently inferred from genetic data[8]. Nowadays, peanut is grown in more than 100 countries, with China being the first for production and India the first for the cultivated area.

*A. hypogaea* is a self-pollinating species characterized by low levels of genetic variation resulting from a series of domestication

[1]Institute of Crop Molecular Breeding, Henan Academy of Agricultural Sciences, Zhengzhou, China. [2]Henan Provincial Key Laboratory for Genetic Improvement of Oil Crops, Zhengzhou, China. [3]National Innovation Centre for Bio-breeding Industry, Xinxiang, China. [4]The Shennong Laboratory, Zhengzhou, China. [5]Department of Soil, Plant and Food Sciences, University of Bari Aldo Moro, Bari, Italy. [6]Plant Breeding, Wageningen University and Research, Wageningen, The Netherlands. [7]Agricultural Genomics Institute at Shenzhen, Chinese Academy of Agricultural Sciences, Shenzhen, China. [8]Sequentia Biotech, Barcelona, Spain. [9]These authors contributed equally: Zheng Zheng, Ziqi Sun, Feiyan Qi, Yuanjin Fang, Ke Lin, Stefano Pavan. ✉e-mail: zhengzheng@hnagri.org.cn; haasz@126.com

bottlenecks[9,10]; nonetheless, it displays large morphological variation. The absence or presence of flowers on the main axis and the flowering pattern, alternate or sequential, are at the basis of the classification of *A. hypogaea* in two subspecies, *A. hypogaea* subsp. *hypogaea* (*Ahh*) and *A. hypogaea* subsp. *fastigiata* (*Ahf*)[11]. Additional traits led to the distinction of two botanical varieties within *Ahh* (var. *hypogaea* and var. *hirsuta*) and four within *Ahf* (var. *fastigiata*, var. *vulgaris*, var. *aequatoriana* and var. *peruviana*)[11]. Breeding resulted in hybridization among these taxa and thus irregular morphologies. Today, a widely used peanut classification is in accordance with five main market types (Virginia, Runner, Peruvian Runner, Valencia and Spanish)[12]. Analysis of genetic structure resulted in clustering patterns approximately in accordance with both classifications[13,14].

Recently, the International Peanut Genome Initiative and two research groups announced the release of cultivated peanut genome assemblies[5,15,16], thus paving the way for in-depth exploration of peanut genetic diversity. Here aiming to define the genetic structure and evolutionary history of peanuts, we performed chloroplast and whole-genome sequencing of peanut accessions belonging to a global peanut collection, encompassing 18 diploid *Arachis* species, *A. monticola* and *A. hypogaea*. Mapping approaches, based on genome-wide association study (GWAS) and recombinant inbred line (RIL) population linkage analysis, were followed to identify candidate genes and genomic regions associated with peanut diversification, domestication and breeding.

## Results

### Sequencing and genotyping

Chloroplast de novo sequencing was performed on 36 wild *Arachis* accessions (34 from diploid wild species and 2 from the tetraploid species *A. monticola*) and a selection of 77 cultivated accessions that, based on the United States of Department of Agriculture (USDA) taxonomic descriptors[17], could be unambiguously assigned to *A. hypogaea* subspecies and botanical varieties (Supplementary Tables 1 and 2). The length of the assembled chloroplast genomes ranged between 156,258 bp and 160,366 bp (Supplementary Table 3). In total, 1,884 polymorphisms (both SNPs and insertions/deletions (InDels)) were found between the 113 assembled chloroplast genomes. Most of the polymorphic sites occurred between wild and cultivated peanuts, whereas 14 polymorphisms were found within *A. hypogaea* (Supplementary Table 4). Eight additional polymorphic sites were found in a panel including, besides *A. hypogaea*, accessions representing six wild species of the AA genome section (Supplementary Table 4). Sanger sequencing and/or kompetitive allele-specific PCR (KASP) assays[18] allowed the validation of five randomly chosen chloroplast polymorphisms detected by de novo sequencing (Supplementary Fig. 1 and Supplementary Table 5). As an independent approach to revealing chloroplast DNA polymorphisms, the 113 assembled chloroplast genomes were processed to identify mononucleotide repeat (MNR) loci, representing the most frequent class of microsatellite loci in chloroplast genomes[19–23]. On average, 10,515 MNR loci were detected across the analyzed genomes (Supplementary Table 6).

Whole-genome resequencing (WGR) was performed on 11 *A. duranensis*, 1 *A. ipaensis*, 2 *A. monticola* and 353 *A. hypogaea* accessions originating from different countries (Fig. 1a and Supplementary Tables 1 and 2), resulting in 160.46 billion reads and 14.54 terabase pairs of clean data. Following alignment against the peanut cv. *Tifrunner* genome assembly[15], unique mapped reads of the 355 tetraploid *A. hypogaea* accessions were associated with 29.00× mean depth and 88.12% genome coverage (Supplementary Table 7). No significant difference was found between the two unique mapped read rates associated with accessions assigned to *Ahh* (88.087%) and *Ahf* (88.220%; Supplementary Table 8). In total, 864,179 SNPs and 71,052 InDels were obtained after quality control. About 40% of the variants were located on the first ten chromosomes (corresponding to the A subgenome), resulting in one variant every 3 kb on average, while 60% of the variants were located on the last ten chromosomes (the B subgenome), resulting in one variant every 2.6 kb on average. The application of KASP assays to a panel of 30 SNP loci and 10,650 data points resulted in the validation of 97.5% of the SNP calls (Supplementary Tables 9 and 10).

### The evolutionary history and genetic structure of peanuts

Chloroplast genomes are maternally inherited; therefore, chloroplast DNA sequences are widely used to infer maternal lineage(s), leading to the origin of allopolyploids[24,25]. Phylogenesis based on chloroplast genome SNPs and InDels indicated *A. duranensis* as the last wild species to diverge before *A. hypogaea*, in accordance with previous studies suggesting *A. duranensis* as the donor of the *A. hypogaea* maternal genome[26]. Remarkably, three *A. duranensis* accessions (PI219823, PI468201 and PI468202), together with one *A. archeri* accession (PI604844) previously shown to be most likely a misclassified *A. duranensis*[27], were included with maximum bootstrap support in a phylogenetic clade-specific for *Ahh*, except for the *Ahf* accessions N524 and N530 (Fig. 1b). Pedigree notes indicated that N524, which was classified as *Ahf* based on morphologic traits, indeed inherited an *Ahh* chloroplast genome (Supplementary Fig. 2). Clustering based on MNR loci also confirmed the presence of two *A. duranensis* accessions (PI219823 and PI475883) in a clade mostly referable to *Ahh* (Supplementary Fig. 3). Both SNP/InDel and MNR-based phylogeneses also provided strong bootstrap support for the occurrence of a clade referable to *Ahf* germplasm, except for N496 (Fig. 1b and Supplementary Fig. 3). Overall, the clear-cut phylogenetic divergence between *Ahh* and *Ahf*, together with grouping of several *A. duranensis* accessions in the *Ahh* intraspecific clade, strongly indicate that different *A. duranensis* mother lineages, and thus allopolyploidization events, originated *Ahh* and *Ahf*.

Although *A. monticola* is thought to be the wild progenitor of *A. hypogaea*, the two *A. monticola* accessions genotyped in this study diverged after the split between *Ahh* and *Ahf*, as they clustered with *Ahh* in the chloroplast phylogenesis (Fig. 1b and Supplementary Fig. 3). This suggests that these two accessions are indeed feral forms originating from *Ahh* hybridization. Further studies, considering more accessions classified as *A. monticola*, might clarify the position of this species in the evolutionary history of peanuts.

Nuclear polymorphism data from the same tetraploid accessions used for chloroplast phylogenesis were also subjected to genetic structure analysis. Principal components analysis (PCA) and maximum likelihood hierarchical clustering provided further support for the clear-cut differentiation between the two *A. hypogaea* subspecies and, within *Ahf*, the botanical varieties *fastigiata*, *vulgaris* and *peruviana* (Fig. 1c,d).

Two additional nuclear trees were obtained for the A and B genomes (Supplementary Fig. 4). Inconsistencies between A genome hierarchical clustering (Supplementary Fig. 4a) and the chloroplast

**Fig. 1 | Peanut phylogenesis and genetic structure. a**, Geographic distribution of 355 *Arachis* accessions resequenced in this study. The color proportion of the circle is proportional to the number of accessions of different types. The map was generated using the mapPies function in the freely available rworldmap R package. **b**, Chloroplast phylogeny obtained by de novo sequencing of 36 wild *Arachis* species and 77 primitive landraces assigned to *A. hypogaea* subspecies and botanical varieties. **c,d**, Results of hierarchical clustering (**c**) and PCA (**d**) from WGR of the same tetraploid accessions in **b**. **e–j**, Polymorphic sites between ($P_B$) and across ($P_A$) two groups of individuals randomly sampled—one from *Ahh* and one from *Ahf* (**e,f**), both from *Ahh* (**g,h**) and both from *Ahf* (**i,j**). Results refer to 100 bootstrap replications and are presented separately for the *A. hypogaea* A genome (chromosomes 1–10) and B genome (chromosomes 11–20). **k**, Extent of LD decay in different *A. hypogaea* botanical varieties and types. **l,m**, PCA (**l**) and parametric clustering (**m**) of the 355 *Arachis* accessions resequenced in this study. mon, monticola; hyp, hypogaea; fas, fastigiata; per, peruviana; vul, vulgaris; hir, hirsuta.

genome phylogenesis (Fig. 1b) can be explained by recombination between homeologous chromosomes, with this event being very common in angiosperm polyploids[28,29]. This would have caused the fixation of DNA segments from the B paternal genome in chromosomes 1–10 of cultivated peanuts. Significantly, homeologous chromosomal rearrangements were reported in *A. hypogaea*, which changed the genomic

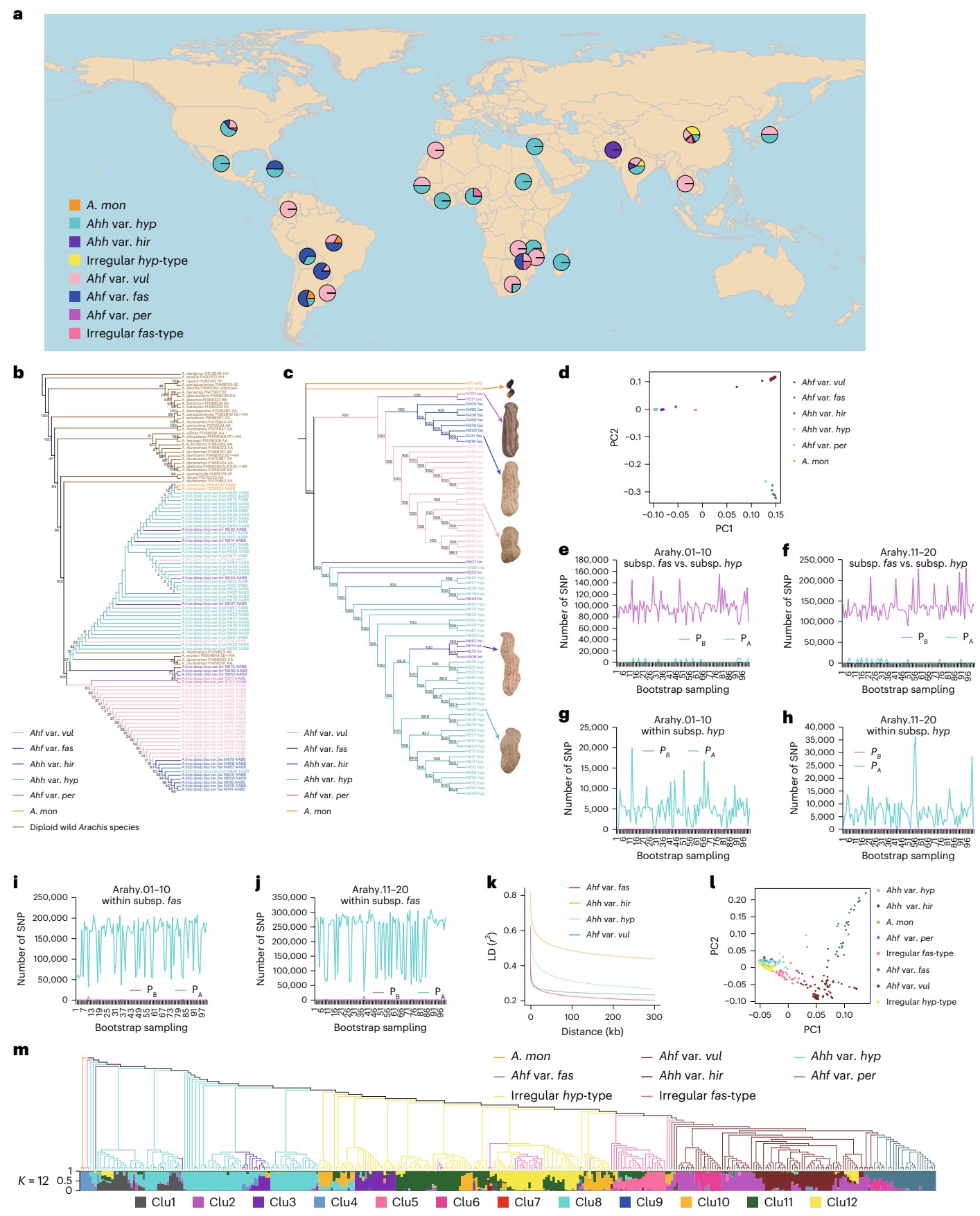

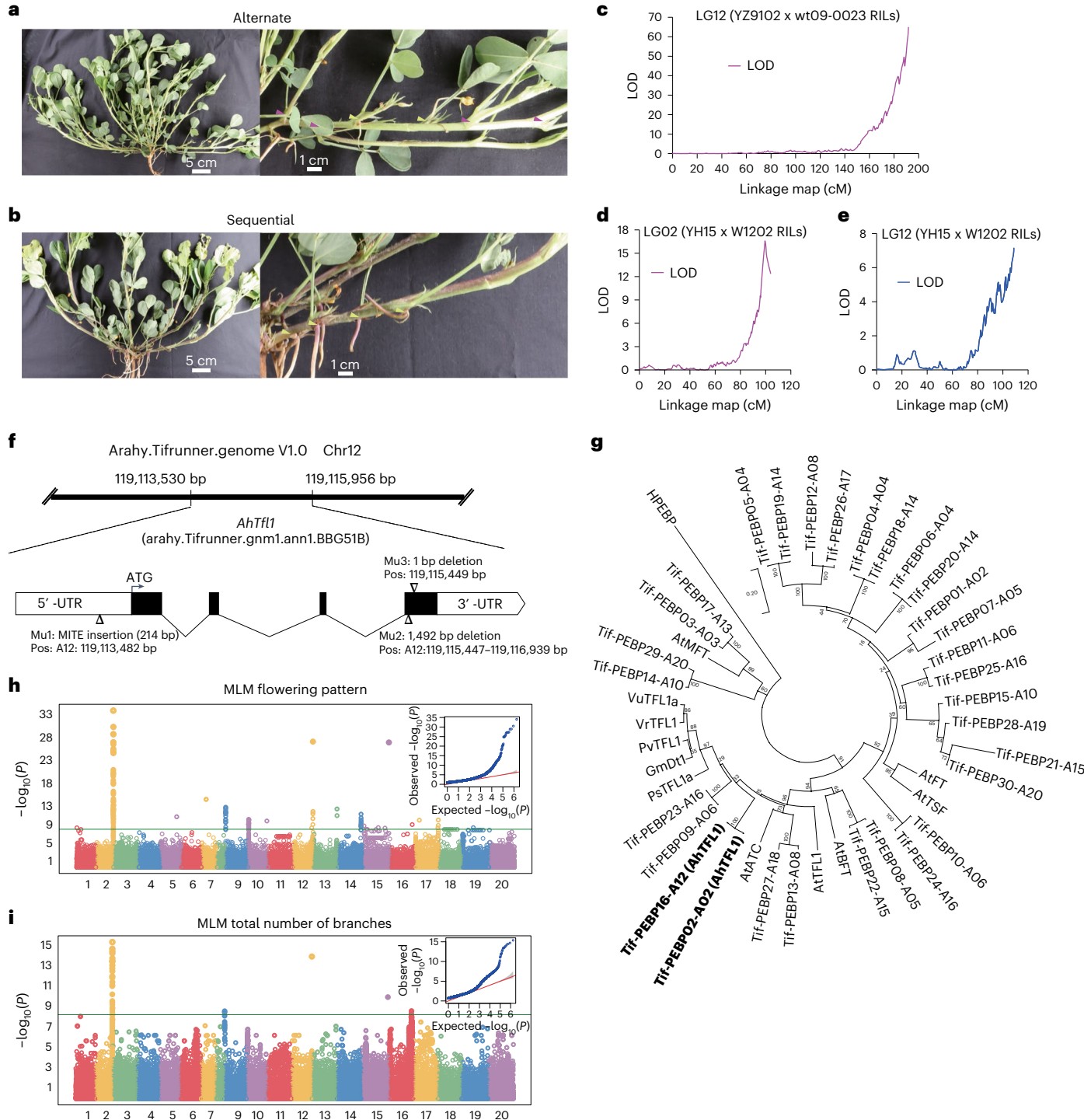

**Fig. 2 | Genetic control of peanut flowering pattern and TNBs. a**, Alternate pattern. **b**, Sequential pattern. **c**–**e**, Likelihood of odd (LOD) score graphs obtained by QTL composite interval mapping using the YZ9102 x wt09-0023 RIL population (**c**) and the YH15 x W1202 RIL population (**d**,**e**). **f**, Structure of the *AhTFL1* gene and features of the three mutations found in the GWAS population. **g**, Phylogenetic relationships among *Arachis* and *Arabidopsis TFL* homologs. *AhTFL1* and *AtTFL1* are highlighted in bold. **h**,**i**, GWAS Manhattan and quantile–quantile (Q–Q) plots for the flowering pattern (**h**) and TNBs (**i**). The MLM implemented in the R package GAPIT was used to test for marker–trait association. The horizontal line in each Manhattan plot indicates the $-\log_{10}P$ threshold for significant association after the Bonferroni correction. The shaded area in the Q–Q plots indicates the 95% confidence interval under the null hypothesis of no association between the SNPs and the trait, under the assumption of a uniform [0, 1] distribution for the *P* values. Mu, mutations.

formula of specific chromosomal regions from the expected AABB to AAAA or BBBB[15,30]. In addition, misassemblies of homeologous regions in the reference genome might also affect nuclear phylogenesis. With this respect, the newly released Tifrunner v2 assembly reports several changes in correspondence of homeologous regions.

Finally, inconsistencies between nuclear and chloroplast tree topologies have been commonly observed in plants[31] and among nuclear peanut phylogenies[32].

The analysis of genomic SNP distribution provided further evidence that different allopolyploids originated *Ahh* and *Ahf*. Indeed,

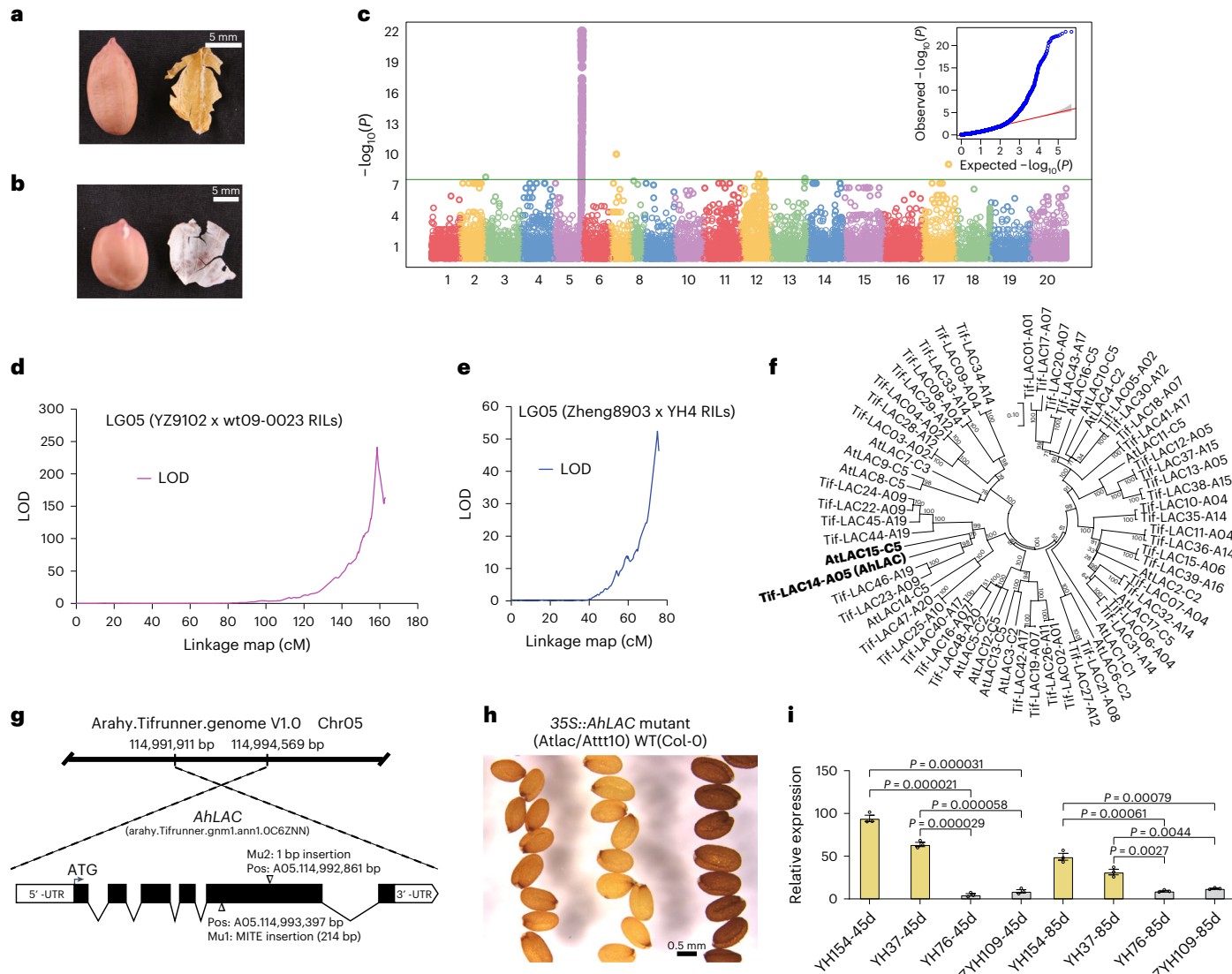

**Fig. 3 | Genetic control of the inner tegument color. a**, Yellow phenotype.
**b**, White phenotype. **c**, GWAS Manhattan plot and Q–Q plot. The MLM implemented
in the R package GAPIT was used to test for marker–trait association. The
horizontal line in the Manhattan plot indicates the $-\log_{10}P$ threshold for
significant association after the Bonferroni correction. The shaded area in
the Q–Q plot indicates the 95% confidence interval under the null hypothesis
of no association between the SNPs and the trait, under the assumption of
a uniform [0, 1] distribution for the $P$ values. **d,e**, Chromosome 5 LOD score
graphs obtained by QTL composite interval mapping using the YZ9102 x
wt09-0023 RIL population (**d**) and the Zheng8903 x YH4 RIL population (**e**).

**f**, Phylogenetic relationships among peanut and *Arabidopsis LAC* homologs.
*AhLAC* and *Arabidopsis TRANSPARENT TESTA 10* (*AtTT10*) are highlighted in bold.
**g**, Structure of the *AhLAC* gene and features of the two mutations found in the
GWAS population. **h**, Complementation of the *Arabidopsis Atlac/Attt10* mutant
with peanut *AhLAC*. The phenotype of the WT Col-0 accession is also shown.
**i**, *AhLAC* expression levels in the seed coat of yellow (YH154 and YH37) and white
tegument (YH76 and ZYH109) accessions at 45 and 85 days after flowering.
Data in **i** are given as mean ± s.e.m.; $n = 3$ biologically independent samples; the
two-tailed Student's *t* test was carried out to compare means. WT, wild type.

bootstrap sampling of groups of individuals from *Ahh* and *Ahf* revealed
a large excess of polymorphisms between groups ($P_B$) compared with
polymorphisms shared across groups ($P_A$), in accordance with a
scenario in which alleles that were polymorphic between different
tetraploid progenitors were fixed in *Ahh* and *Ahf* (Fig. 1e,f). In contrast,
sampling of group pairs within the same subspecies yielded opposite
results ($P_B \ll P_A$; Fig. 1g–j), in agreement with their descendance from
a common tetraploid progenitor. With a few exceptions, we found
a roughly even distribution of the $P_A$ and $P_B$ polymorphism classes in
the genome (Supplementary Fig. 5).

Linkage disequilibrium (LD) decay significantly varied within
*A. hypogaea*, as it was slower in var. *hirsuta* and *hypogaea* than in var.
*fastigiata* and *vulgaris* (Fig. 1k), which is consistent with the lower
level of genetic diversity found in var. *hirsuta* and var. *hypogaea*

(Supplementary Fig. 6). The half-maximum decay distance was 99.4 kb
within var. *hypogaea*, 174.5 kb within var. *hirsuta*, 5.6 kb within var.
*fastigiata* and 15.8 kb within var. *vulgaris*.

To identify genomic regions that are highly divergent between
the peanut subspecies *Ahh* and *Ahf*, thus contributing to their diver-
sification, we estimated haplotypes and found specific haplotypes
distinguishing the botanical varieties (Supplementary Fig. 7).

The effect of the recent breeding history on peanut genetic
structure was investigated using the whole panel of 355 accessions
sequenced in this study, also including cultivars derived from hybridi-
zation breeding programs. Parametric modeling, PCA and hierarchical
clustering (Fig. 1l,m and Supplementary Table 11) defined additional
levels of population stratification. In more detail, within var. *hypogaea*,
one cluster was associated with several Chinese landraces (Cls8) and

one (Cls1) with American varieties or derivatives. Within var. *vulgaris*, distinct clusters were found for African landraces (Cls6), Chinese landraces (Cls2) and cultivars from southern China (Cls7). Cls9 was found mainly for var. *fastigiata*. Finally, five clusters (Cls3, Cls5, Cls10, Cls11 and Cls12) were found for irregular-type peanuts, originating from hybridization between the two *A. hypogaea* subspecies, with Cls3 and Cls5 being morphologically more similar to *Ahh* and *Ahf*, respectively.

## Genes associated with divergence between peanut subspecies

Different evolutionary histories of the peanut subspecies *Ahh* and *Ahf* were accompanied by the fixation of contrasting phenotypes for several traits, including the flowering pattern, the number of branches, the growth habit and the color of the inner seed tegument. The flowering pattern, sequential in *Ahf* and alternate in *Ahh* (Fig. 2a,b), is thought to have a major role in the adaptation to different ecosystems. Mapping this trait by two RIL populations originating from different parental lines identified, in one case, a major Quantitative Trait Locus (QTL) at the end of chromosome 12 and, in the other, two QTLs at the end of chromosomes 2 and 12 (Fig. 2c–e and Supplementary Table 12). Recombinant screening using a set of newly developed KASP markers allowed us to fine-map the QTL on chromosome 12 in a 514.83 kb region containing 52 genes (Supplementary Fig. 8 and Supplementary Table 13). Among them, a gene (*arahy.BBG51B*) encoding a phosphatidylethanolamine-binding protein was the only one associated with a frameshift mutation (Supplementary Table 13). Notably, based on phylogenetic reconstruction, this gene, named *AhTFL1*, was deemed as the putative orthologue of *AtTFL1* (*AT5G03840*), involved in the control of inflorescence architecture in *Arabidopsis*[33–36] (Fig. 2f,g and Supplementary Table 14), thus making *AhTFL1* an obvious candidate to control the flowering pattern in peanut. GWAS confirmed the presence of strong signals for markers closely associated with *AhTFL1* on the terminal regions of chromosome 2 (377.48 kb, $-\log_{10}P = 34.05$) and chromosome 12 (14.4 kb, $-\log_{10}P = 27.31$; Fig. 2h and Supplementary Table 15), suggesting that *AhTFL1* homologs on the A and B genomes are both contributing to the flowering pattern phenotype. *AhTFL1* sequencing in the GWAS population revealed the occurrence of three mutations (a MITE insertion, a 1,492 bp deletion and a 1 bp deletion; Fig. 2f and Supplementary Fig. 9) fully cosegregating with the sequential flowering pattern (Supplementary Tables 16 and 17). Notably, a recent work[37] also reports full cosegregation between the MITE InDel described in our study and the peanut flowering pattern, as well as significantly lower expression of *AhTFL1* in (1) *Ahf* compared with *Ahh* and (2) flowering compared with non-flowering branches. GWAS for the total number of branches (TNBs) resulted in the strongest signal colocalizing with *AhTFL1*, indicating that *AhTFL1* may have a pleiotropic effect on this trait (Fig. 2i and Supplementary Table 18).

Another trait displaying divergent phenotypes between the two peanut subspecies is the color of the seed inner tegument, which is invariably yellow in *Ahh* (Fig. 3a) and white in *Ahf* (Fig. 3b). Both GWAS and RIL-based mapping highlighted the strong association between the tegument color and a genomic region on chromosome 5 (Fig. 3c–e and Supplementary Tables 12 and 19). Screening of recombinant RILs by KASP markers allowed to fine-map the QTL to an interval of 540.14 kb (Supplementary Fig. 10a), which was further refined to 107.88 kb by the screening of 7,900 segregant F$_2$ individuals (Supplementary Fig. 10b). Within the interval, a gene (*arahy.OC6ZNN*) encoding a laccase-like protein, named *AhLAC*, was the only one associated with a

frameshift mutation (Supplementary Table 20). Notably, this gene is the putative ortholog of the *Arabidopsis* gene *AtLAC15* (also referred to as *TRANSPARENT TESTA 10* or *AtTT10, AT5G48100*; Fig. 3f and Supplementary Table 21), which was shown to influence the color of the seed coat through its enzymatic role in the oxidative polymerization of flavonoids[38]. The strongest GWAS signal ($-\log_{10}P = 22.17$) was 68.96 kb from *AhLAC* (Fig. 3c and Supplementary Table 19). *AhLAC* sequencing in the GWAS population revealed the occurrence of two mutations (a MITE insertion and a 1 bp insertion, the latter only occurring in the two *Ahf* var. *peruviana* accessions; Fig. 3g and Supplementary Fig. 11). A KASP assay was designed on the MITE insertion (Supplementary Table 22) and verified to be fully cosegregating with the inner tegument color in both the GWAS population and the YZ9102 x wt09-0023 RIL population (Supplementary Tables 23–25). Heterologous overexpression of *AhLAC* partially complemented the *Arabidopsis Attt10* loss-of-function mutant in four independent transgenic lines, with the level of seed lightness (expressed by the L* score) being inversely related to the transgene expression level (Fig. 3h and Supplementary Fig. 12a–c). Consistently, the yellow inner tegument accessions YH154 and YH37 displayed markedly higher *AhLAC* expression than the white inner tegument accessions YH76 and ZYH109 (Fig. 3i). Finally, the epicatechin content was significantly higher in YH76 and ZYH109 than in YH154 and YH37 (Supplementary Fig. 12d,e), consistent with the possibility that *AhLAC* causes pigmentation through epicatechin oxidative polymerization, similarly to Arabidopsis *AtTT10* (ref. 38).

## Genetic dissection of main peanut economic traits

The peanut growth habit (erect or prostrate; Fig. 4a,b) strongly conditions cultivation practices[39]. Both GWAS and genetic mapping using two RIL populations resulted in signals for a genomic region on chromosome 15 (Fig. 4c–e and Supplementary Tables 12 and 26), in accordance with previous studies[40–42]. Recombinant screening using a set of newly developed KASP markers allowed to fine-map the QTL on a 299.11 kb region containing 20 genes (Supplementary Fig. 13 and Supplementary Table 27). Among them, a *MADS-box* gene (*arahy.ATH5WE*) was chosen as a candidate, as (1) it was the only one displaying a mutation within the coding sequence (a frameshift caused by a 1,870 bp deletion), and (2) the *MADS-box* family of transcription factors was previously associated with the plant growth habit[43]. The most significant GWAS signal on chromosome 15 ($-\log_{10}P = 9.12$) was only 2.07 kb apart from the same *MADS-box* homolog, which, based on phylogenetic analysis, was related to *Arabidopsis AtPI* (*At5G20240*) and *AtAP3* (*At3G54340*; Fig. 4g and Supplementary Table 28). Three gene mutations (a 2 bp insertion in the first exon, a 1,870 bp deletion in the first intron and a MITE insertion in the third intron) were characterized (Fig. 4f and Supplementary Fig. 14), and the polymorphisms were used to develop KASP and Integrative Genomics Viewer markers. With a few exceptions, at least one of the three mutations was found to cosegregate with the erect phenotype (Fig. 4h,i) in the GWAS population (Supplementary Tables 29 and 30). Considering that the growth habit might be influenced by environmental factors, further investigations are required to clarify the putative role of a *MADS transcription factor* as a determinant of the peanut growth habit.

Pod and kernel dimensions, together with kernel oil content, are key peanut commercial traits. RIL-based mapping indicated that kernel weight, kernel length and pod weight are genetically correlated. The identification of QTLs on chromosomes 5 and 16 is in accordance

---

**Fig. 4 | Genetic control of the growth habit. a**, Erect phenotype. **b**, Prostrate phenotype. **c**, GWAS Manhattan plot and Q–Q plot. The MLM implemented in the R package GAPIT was used to test for marker–trait association. The horizontal line in the Manhattan plot indicates the threshold for significant association after the Bonferroni correction. The shaded area in the Q–Q plot indicates the 95% confidence interval under the null hypothesis of no association between the SNPs and the trait, under the assumption of a uniform [0, 1] distribution for

the *P* values. **d,e**, Chromosome 15 LOD score graphs obtained by QTL composite interval mapping using the YZ9102 x wt09-0023 RIL population (**d**) and the YH15 x W1202 RIL population (**e**). **f**, Structure of the *AhMADS-box* transcription factor 6 gene and features of the three mutations found in the GWAS population. **g**, Phylogenetic relationships among *Arachis* and *Arabidopsis AGL* homologs. *AtPI, AtAP3* and *AhMADS-box* are highlighted in bold. **h,i**, The distribution of 353 accessions according to growth habit (**h**) and mutation types (**i**).

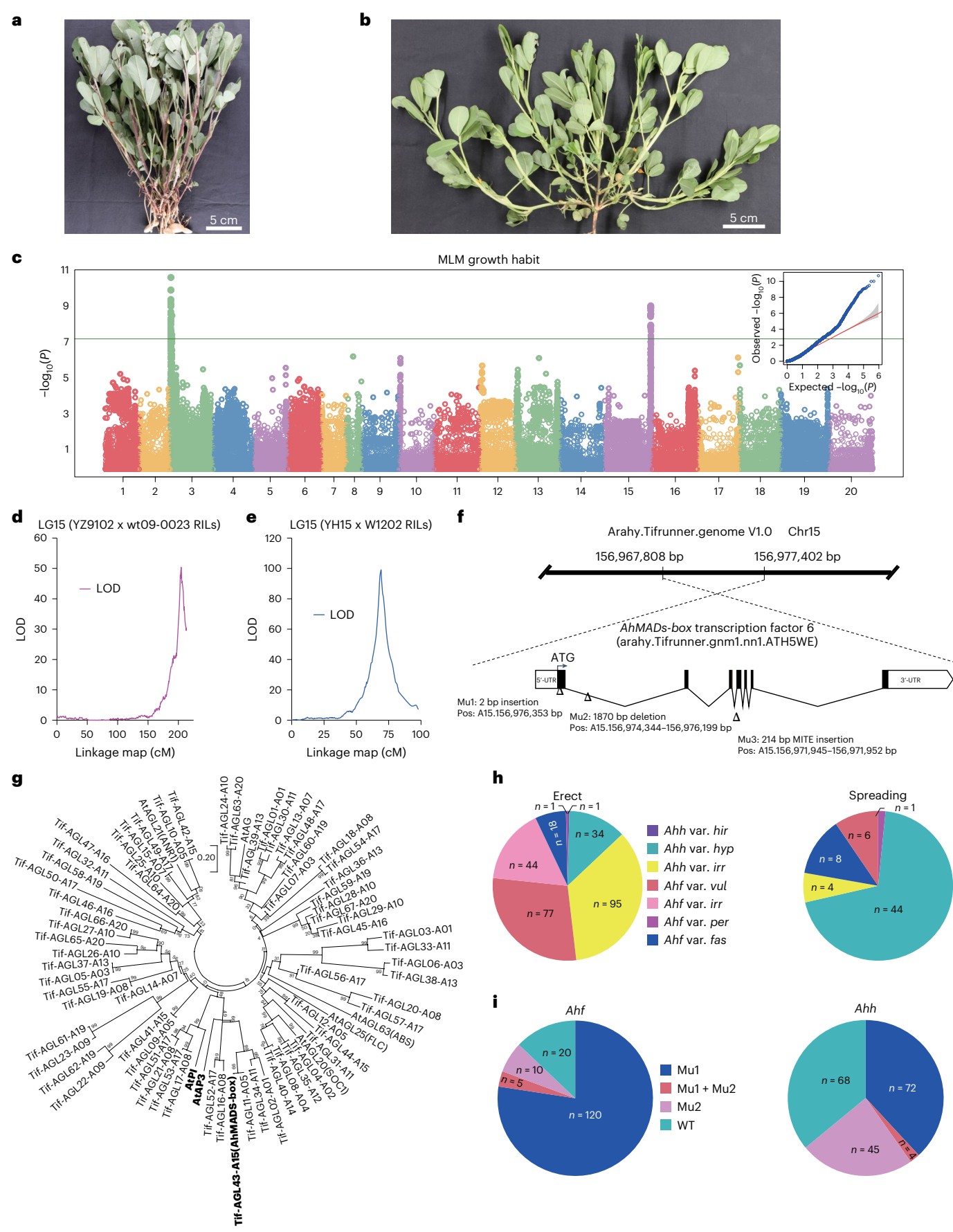

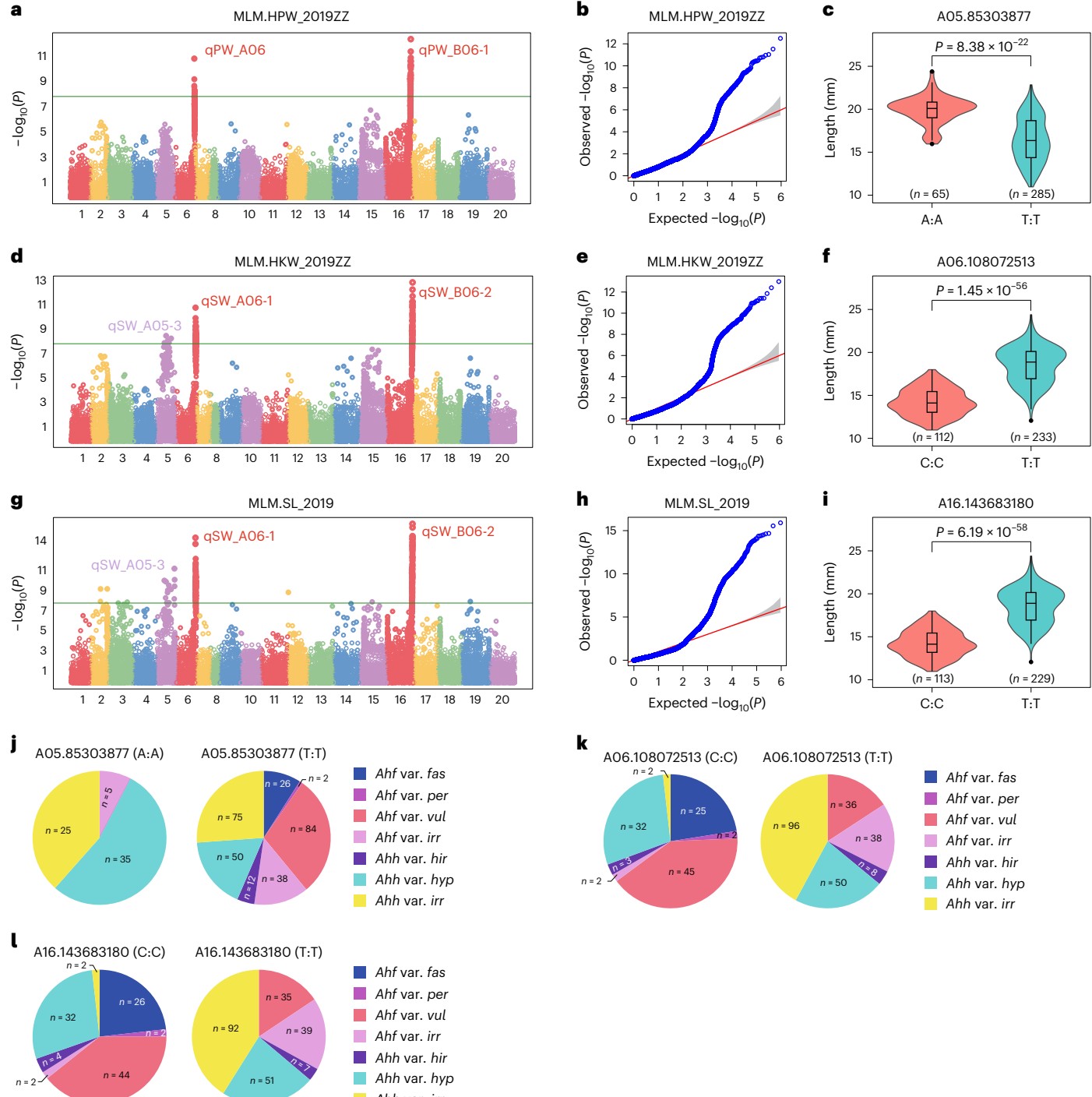

**Fig. 5 | GWAS for HPW, HKW and SL. a,d,g**, Manhattan plots for HPW (**a**), HKW (**d**) and SL (**g**). The MLM implemented in the R package GAPIT was used to test for marker–trait association. The horizontal line in each Manhattan plot indicates the −log₁₀$P$ threshold for significant association after the Bonferroni correction. **b,e,h**, Q–Q plots for HPW (**b**), HKW (**e**) and SL (**h**). The shaded area in each Q–Q plot indicates the 95% confidence interval under the null hypothesis of no association between the SNPs and the trait, under the assumption of a uniform [0, 1] distribution for the $P$ values. **c,f,i**, Violin plots and boxplots describing the SL distribution in different accessions with genotypes AA ($n = 65$) and TT ($n = 285$) at the A05.85303877 locus (**c**), genotypes CC ($n = 112$) and TT ($n = 233$) at the A06.108072513 locus (**f**), and genotypes CC ($n = 113$) and TT ($n = 229$) at the A16.143683180 locus (**i**). In the boxplots, centerline indicates the median; box lower and upper edges indicate the 25% and 75% quartiles, respectively; whiskers indicate 1.5× interquartile range; the two-tailed Student's $t$ test was carried out to compare means. **j,k,l**, Pie-charts for the proportion of accessions falling in different *A. hypogaea* types and having alternative genotypes at the sites A05.85303877 (**j**), A06.108072513 (**k**) and A16.143683180 (**l**).

with previous studies[44,45]. GWAS confirmed marker–trait associations on chromosomes 5 and 16 (significance peaks for −log₁₀$P$ = 13.05 and 15.91, respectively); however, a signal on chromosome 6 was also found (Fig. 5a,d,g and Supplementary Table 31). Finally, GWAS for oil

content highlighted a main signal on chromosome 8 (−log₁₀$P$ = 8.94; Supplementary Fig. 15 and Supplementary Table 32), in correspondence with a previously mapped QTL[46] and in accordance with the recent findings in ref. 8.

## Discussion

This study reports the results of a massive DNA sequencing effort, allowing the fine-scale reconstruction of main events associated with the evolutionary history and phenotypic diversification of peanuts. Chloroplast genome sequencing and phylogenesis from a large germplasm panel, including several accessions of the chloroplast donor wild species *A. duranensis*, provided a solid indication that the peanut subspecies *Ahh* and *Ahf* result from distinct polyploidization and domestication events. This was confirmed by the characterization of genetic polymorphisms between and within the two taxonomic groups. Notably, multiple polyploidization events were reported to be at the basis of the evolution of several species[25,47–49], in accordance with our findings. The independent origin of *Ahh* and *Ahf* explains the contradictory findings from ref. [5] and refs. [15,30], tracing back peanut polyploidization <10,000 years ago and 0.42–0.47 million years ago, respectively, which were previously debated[50,51]. Indeed, the two research groups based their evolutionary analyses on different reference genome sequences, from the *Ahh* cultivar Tifrunner and the *Ahf* cultivar Shitouqi. We predict that research methods used in this study might be transferred to other allopolyploid plant species whose origin is still elusive.

Two independent mapping approaches (GWAS and biparental linkage analysis) were used to investigate the genetic basis of phenotypic divergence between *Ahh* and *Ahf* and the genetic control of several economically important traits. This choice is justified by the need to increase the confidence of the results obtained by the GWAS approach that, while allowing a higher mapping resolution, might lead to false positive signals in the case of A and B genome homeologous regions showing high sequence similarity or in the case of homeologous recombination changing the genomic formula from the expected AABB to AAAA or BBBB[15,30]. The identification of *AhTFL1* as the gene putatively controlling the peanut flowering pattern is in line with previous findings in *Arabidopsis* and other crop species, although the peanut raceme inflorescence bears distinctive features. Continuous flowering often coincides with early maturing, which is desirable in areas characterized by a shorter growing season. Notably, we also showed that the pigmentation of the inner tegument likely originates from a mutation of the laccase gene *AhLAC*, although further investigation is needed to clarify whether *AhLAC* promotes tegument pigmentation through the oxidative polymerization of flavonoids, as it was shown for its *Arabidopsis* homolog *AtTT10* (ref. [38]). The tegument color can affect several physiologic and economic traits in plants, including legumes, such as seed dormancy, response to pathogens and pests and seed nutritional traits[52,53]; thus, this finding can be of broad interest for plant scientists and breeders. It might be speculated that the occurrence of white inner tegument in *Ahf* contributes to the absence of seed dormancy in this subspecies, in contrast with *Ahh*. This trait makes *Ahf* more suitable for cultivation in warm environments and allows consecutive harvests.

Together, the data reported in this study provide an important genomic resource for further and faster peanut genetic improvement, and the results presented here might be of broad interest to the plant sciences community and plant breeding.

## Online content

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

# Methods

## Plant material and DNA extraction

The germplasm panel used in this study included 34 accessions of wild diploid species, 2 accessions of wild tetraploid *A. monticola*, 353 accessions of cultivated tetraploid *A. hypogaea* and three previously described RIL populations of *A. hypogaea* (Supplementary Tables 1 and 2)[54-75]. The 353 *A. hypogaea* accessions were selected from more than 2,000 accessions collected from 27 countries and 18 China provinces based on tunable genotyping-by-sequencing (tGBS) sequencing and phenotype cluster analysis[13]. The diversity panel included five peanut botanical varieties (85 var. *hypogaea*, 12 var. *hirsuta*, 26 var. *fastigiata*, 84 var. *vulgaris* and 2 var. *peruviana*) and two kinds of irregular types (100 irregular *hypogaea*-type and 44 irregular *fastigiata*-type) associated with landraces, cultivars and breeding lines. Genomic DNA extraction was performed on the whole germplasm panel using the Plant Genomic DNA Kit (Tiangen Biotech).

## Chloroplast de novo sequencing and variant identification

In total, 113 chloroplast genomes (77 from *A. hypogea* landraces representing var. *hypogaea*, var. *hirsuta*, var. *fastigiata*, var. *vulgaris* and var. *peruviana;* 2 from *A. monticola* accession; and 34 from wild diploid accessions) were de novo assembled using default pipeline settings (-R 15 -k 21,45,65,85,105) of the GetOrganelle toolkit version 1.7.3.5 (ref. 76). The chloroplast genomes and repeat_pattern1 that consist of two equimolar isomeric sequences and with the same direction of the small single-copy (SSC) regions were used for making alignments with the MAFFT program version 7 (ref. 77) for pairwise comparisons. The SNP and InDel variants between the chloroplast genomes were identified using MEGA X[78] with the Chlorophycean Mitochondrial code set.

## Genomic resequencing and variant identification

Paired-end DNA libraries with inserts of approximately 300 bp were constructed and sequenced using the Illumina HiSeq Xten (Illumina) platform with PE151. Raw data were cut with an average coverage of 20× per sample for further analysis. High-quality reads passing the quality check and filtering were aligned to the genome of cultivated peanut *A. hypogaea* cv. Tifrunner version 1 using minimap2 (v2.10)[79] with the command '-ax sr -t 25 -K 5G'. BAM alignment files were then generated with sambamba (v0.6.8)[80] by removing potential PCR duplications.

SNP and InDel calling were performed with the Genome Analysis Toolkit (v4.0.12.0)[81] with the HaplotypeCaller method. Detected SNPs matching any of the following conditions were filtered out: QualByDepth <2.0, FisherStrand >60.0, RMSMappingQuality <40.0, MappingQualityRankSumTest <−12.5 and ReadPosRankSumTest <−8.0. The conditions used to filter out InDels were as follows: QualByDepth <2.0, FisherStrand >200.0 and ReadPosRankSumTest <−20.0. After applying the aforementioned filtering conditions, we obtained variationSet1. To further exclude variant calling errors, all variations with a missing rate >0.05 (alleles having less than five reads supporting them were marked as missing), minor allele frequency <0.01 and number of heterozygous genotypes >10 were filtered out using vcftools (v0.1.19)[82] and bcftools (v1.10.2)[83], which resulted in variationSet2.

## Chloroplast phylogenesis

The 113 chloroplast genomes were configured, of which the SSC regions aligned in the same direction were used to construct the neighbor-joining tree with MEGA X[78].

The multi-FASTA file containing the 113 assembled chloroplast genomes was analyzed with an in-house generated Python script to identify and count mononucleotide microsatellites. For each sequence entry in the FASTA file, the script identifies and counts occurrences of monorepetitions of the four nucleotide bases (A, T, G and C) that fall within the length range of 3–20 nucleotides. After counting these repeats, the script calculates the percentage abundance of each SSR type relative to the total sequence length. In the following step, the microsatellites that were not present in any of the samples were discarded, as well as those with the same abundance across all the samples. The obtained matrix of abundance was processed in R to generate an Euclidean distance matrix. Samples were clustered by hierarchical clustering based on the Pearson correlation of the distance values. Bootstrap values were obtained using the R package pvclust using 1,000 iterations.

## Genomic distribution of SNPs between and across subspecies

Two groups of five individuals, one from *Ahh* and the other from *Ahf*, or both from the same subspecies, were extracted by performing 100 bootstrap replicates. SNP data were used to extract polymorphisms between groups ($P_B$), occurring when alternative alleles are fixed in each group (that is, $F_{ST} = 1$), and polymorphisms shared across groups ($P_A$), occurring when alternative alleles are present in both groups. Chromosomes 1–10 and 11–20 were analyzed separately. The density distribution of polymorphisms in 1 Mb genome windows was drawn using the R package Cmplot.

## LD and haplotype block analyses

LD decay was calculated for all pairs of variations on var. *hypogaea* and irregular *hypogaea*-type (183 samples), var. *hirsuta* (12 samples), var. *fastigiata* (26 samples), var. *vulgaris* and irregular *fastigiata*-type (130 samples) from variationSet2 using PopLDdecay (v3.31) with default parameters[84]. Considering the influence of the different number of samples in LD decay calculation, we standardized the sample size of var. *hypogaea* and irregular *hypogaea*-type and var. *vulgaris* and irregular *fastigiata*-type to 12 and 26, respectively, using shuf (version 8.22) and repeated 100 times. Half-maximum decay distance was calculated based on averaging the $r^2$ values of each 100-bp separation bin (that is, average $r^2$ for SNPs separated by 1–100 bp, 101–200 bp, etc.). To calculate the half-maximum decay distance for var. *hypogaea* and var. *vulgaris*, all 100 standardized sample lists were used to calculate the half-maximum decay distance individually, and then the median value was taken (using the built-in quantile function in R with $P = 0.5$ and type = 1). To call haplotype blocks in 79 selected landraces, we used the R package HaploBlocker (v1.5.18)[85] with adaptive mode and different subspecies as subgroups on variationSet2. All 79 samples were clustered with the binary matrix output from haplotype blocks using ade4 in R (v1.7-16)[86] on the first ten chromosomes (subgenome A) and the second ten chromosomes (subgenome B) separately.

## Population structure analysis

After clumping the remaining variants in variationSet2 using PLINK (v1.90b6.9)[87] with '--clump-p11 --clump-p21 --clump-r2 0.5', variations (variationSet3) were retained for population structure analysis. A maximum likelihood phylogenetic tree was constructed with IQ-TREE (v1.6.12)[88] using the optimal model (GTR + F + ASC + R5) as determined by the Bayesian information criterion. Population structure was also studied using ADMIXTURE (v1.30)[89] with *k* between 1 and 20. The program smartpca from the Eigenstrat package (v7.2.1)[90] was used to calculate eigenvectors of variationSet2. Allelic differentiation between populations was measured by nucleotide diversity ($\pi$) of each subspecies group using vcftools (v0.1.19) with a 200 kb window and a step size of 100 kb for each subspecies on variationSet2.

## QTL mapping and GWAS

The three RIL populations YZ9102 x wt09-0023, YH15 x W1202 and Zheng8903 x YH4, including 521, 318 and 212 lines, respectively, were used for QTL mapping. The YZ9102 x wt09-0023 RIL population was sequenced using the single digest restriction site-associated DNA sequencing protocol[91], and the sequencing depths for the two parents and the RILs were approximately 25× and 5×, respectively[75]. SNP sites were used for genetic map construction and QTL mapping as previously

reported[75]. The other two populations were sequenced using WGR, and the sequencing depths for the parents and the RILs were approximately 30× and 1×, respectively[73,74]. The sliding-window approach for genotype calling and recombination breakpoint determination[92] was applied to convert SNPs into bin markers. The genetic maps of the YZ9102 x wt09-0023 and the Zheng8903 x YH4 populations were constructed using Joinmap (v5.0)[93], whereas the genetic map of the YH15 x W1202 population was constructed using QTL Icimapping (v4.2)[94]. QTL analysis was performed using the multiple QTL mapping algorithm implemented in MapQTL (v6.0)[95] by setting the mapping step size as 0.1 cM and the LOD threshold as 2.5. Fine mapping was performed by developing KASP markers in the QTL interval and screening recombinant RILs. As for the inner tegument color trait, further fine mapping was performed by screening recombinant individuals from an $F_2$ population originating from the P573 x P602 cross.

GWAS was carried out on the 353 cultivated peanuts from variationSet2. Phenotypic data for flowering pattern, TNBs, color of the inner tegument, growth habit and oil content based on gas chromatography were collected in one environment (2019: Zhengzhou (2019ZZ)), whereas 100 kernel weight (HKW), 100 pod weight (HPW) and seed length (SL) were collected in seven environments (2017: Yuanyang (2017YY); 2018: Yuanyang (2018YY), Xinyang (2018XY), Weifang (2018WF); 2019: Zhengzhou (2019ZZ), Shangqiu (2019SQ), Weifang (2019WF)) using a randomized complete block design with two replicates. The mixed linear model (MLM)[96] implemented in the R package GAPIT (v3.0)[97] was used to identify significant associations (Supplementary Table 33), using population structure results from ADMIXTURE analysis ($K$), the first two principal components (PCs) and the flowering pattern as covariates. The genome-wide significance threshold for association was set as $0.05/n$ (where $n$ is the number of markers). Significant SNPs in the candidate intervals were annotated using software snpEff (v4.5)[98].

### AhLAC functional characterization

A 35S overexpression vector (PBI121) was constructed by double digestion (XbaI and SacI) and ligation of the AhLAC gene into the vector. The Arabidopsis tt10 mutant (cs2105589) was transformed by Agrobacterium inflorescence immersion[99], and mature seeds were collected. Transgenic-resistant plants were subsequently screened on Murashige and Skoog (MS) medium containing 50 mg l$^{-1}$ kanamycin. The AhLAC gene expression level in transgenic plants was determined by real-time qPCR using the primer pair 5′-ATGAAATGTTGTTGCTTGG-3′ (F)/5′-TCAACAAGGAGGCAGATCTG-3′ (R) in combination with the primer pair 5′-TCCGGACCAGCGTCTCA-3′ (F)/5′-CCACCACGAAGACGCAGGA-3′ (R), the latter targeting the AtUBQ10 housekeeping gene[100]. The level of seed lightness (the L* score) was quantified on a 0–100 scale by the high-precision spectrophotometer NR110 (3nh).

To investigate the functional role of AhLAC in peanut, seed coat AhLAC expression levels were quantified at 45 and 85 days after flowering in peanut genotypes displaying yellow (YH154 and YH37) or white (YH76 and ZYH109) inner teguments. Real-time qPCR was performed using the primer pair 5′-CATGGAGTGAAGCAGCCAAGAA-3′ (F)/5′-AGTGGCTCTTGCCCAATCACT-3′ (R) in combination with the primer pair 5′-GACGCTTGGCGAGATCAACA-3′ (F)/5′-AACCGGACAAC CACCACATG-3′ (R); the latter targeting the ADH3 housekeeping gene[101]. Epicatechin was extracted from the dry seed coat of YH154, YH37, YH76 and ZHY109 using the standard procedure[102] and quantified by the high-performance liquid chromatograph series 6420A mass spectrometry AGILENT 1260 (Agilent Technologies).

### Statistical testing

A two-tailed Student's $t$ test was conducted in basic R (v4.1.3) to compare means relative to unique mapped read rates, relative AhLAC expression, epicatechin content, SL and seed lightness.

### Map generation

The map depicting the country of origin for the peanut accessions considered in this study was generated using the mapPies function in the freely available package rworldmap[103] in R (v4.1.3).

### Reporting summary

Further information on research design is available in the Nature Portfolio Reporting Summary linked to this article.

### Data availability

The datasets analyzed or generated by this study are available in Supplementary Information and the public repositories of the National Center for Biotechnology Information (NCBI, https://www.ncbi.nlm. nih.gov) and Zenodo (https://zenodo.org/). WGR data are available in the NCBI Sequence Read Archive database (Bioproject PRJNA605106). The assembled chloroplast genomes obtained in this study are available in the NCBI GenBank database (accessions from PP971404 to PP971516). Genomic SNPs and InDels identified in this study are available at the Zenodo repository (https://doi.org/10.5281/zenodo.12475904)[104].

### Code availability

The custom script used to extract polymorphisms between and across A. hypogaea subspecies Ahh and Ahf is available at the Zenodo repository (https://doi.org/10.5281/zenodo.12614808)[105]. The in-house generated Python scripts used to count SSRs, calculate their percentage and perform clustering based on SSR data are available at the Zenodo repository (https://doi.org/10.5281/zenodo.12191309)[106].

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

## Acknowledgements

We thank the financial support from the National Key Research and Development Program (2023YFD1202800 to X.Z.), Special Project for National Supercomputing Zhengzhou Center Innovation Ecosystem Construction (201400210600 to Z. Zheng), Henan Provincial R&D Projects of Inter-regional Cooperation for Local Scientific and Technological Development Guided by Central Government (YDZX20214100004191 to B.H.), Major Science and Technology Projects of Henan Province (201300111000 and 221100110300 to X.Z.), China Agriculture Research System of Ministry of Finance People's Republic of China (MOF) and Ministry of Agriculture and Rural Affairs (MARA) (CARS-13 to X.Z.), Henan Provincial Agriculture Research System, China (S2012-5 to W.D.), the Thousand Top Talent Youth in Zhongyuan (ZYQR201912171 to Z. Zheng) and Key Research Project of the Shennong Laboratory (SN01-2022-03 to X.Z.). The work from S. Pavan was partially carried out within the framework of the Agritech

National Research Center, receiving funding from the European Union Next-Generation EU (PIANO NAZIONALE DI RIPRESA E RESILIENZA (PNRR)—MISSIONE 4 COMPONENTE 2, INVESTIMENTO 1.4—D.D. 1032 (17 June 2022), CN00000022 to S.P.). We would also like to express our gratitude to the Oil Crops Research Institute, Chinese Academy of Agricultural Sciences, Shandong Peanut Research Institute, and other research institutions for providing peanut germplasms for this study.

## Author contributions

Z. Zheng designed the experiments and wrote the paper. Z.S. and F.Q. prepared the DNA, performed field experiments and analyzed the candidate genes. Y.F. and K.L. analyzed genetic variation and performed GWAS. S.P. contributed to experimental design, data analysis and interpretation, and paper preparation and revision. B.H. and W.D. provided help to design the experiments. P.D. provided the wild accessions. M.T., L.S., J.X., S.H., H.L., L.Q., Z. Zhang, X.D., L.M., R.Z. and J.W. provided help in laboratory and field experiments. A.L., J.R., Y.L., C.M., C.D. and R.A.C. contributed to the analysis of genetic polymorphisms between and within subspecies. Y.B. and R.G.F.V. revised the paper and offered suggestions. X.Z. conceived and facilitated the project, developed the RIL populations and revised the paper. All authors read and approved the final paper.

## Competing interests

The authors declare no competing interests.

## Additional information

**Correspondence and requests for materials** should be addressed to Zheng Zheng or Xinyou Zhang.

# Reporting Summary

## Statistics

For all statistical analyses, confirm that the following items are present in the figure legend, table legend, main text, or Methods section.

| n/a | Confirmed | |
|---|---|---|
| ☐ | ☒ | The exact sample size (*n*) for each experimental group/condition, given as a discrete number and unit of measurement |
| ☐ | ☒ | A statement on whether measurements were taken from distinct samples or whether the same sample was measured repeatedly |
| ☐ | ☒ | The statistical test(s) used AND whether they are one- or two-sided *Only common tests should be described solely by name; describe more complex techniques in the Methods section.* |
| ☒ | ☐ | A description of all covariates tested |
| ☐ | ☒ | A description of any assumptions or corrections, such as tests of normality and adjustment for multiple comparisons |
| ☐ | ☒ | A full description of the statistical parameters including central tendency (e.g. means) or other basic estimates (e.g. regression coefficient) AND variation (e.g. standard deviation) or associated estimates of uncertainty (e.g. confidence intervals) |
| ☐ | ☒ | For null hypothesis testing, the test statistic (e.g. *F*, *t*, *r*) with confidence intervals, effect sizes, degrees of freedom and *P* value noted *Give P values as exact values whenever suitable.* |
| ☒ | ☐ | For Bayesian analysis, information on the choice of priors and Markov chain Monte Carlo settings |
| ☒ | ☐ | For hierarchical and complex designs, identification of the appropriate level for tests and full reporting of outcomes |
| ☒ | ☐ | Estimates of effect sizes (e.g. Cohen's *d*, Pearson's *r*), indicating how they were calculated |

*Our web collection on statistics for biologists contains articles on many of the points above.*

## Software and code

Policy information about availability of computer code

| Data collection | Sequencing data used for this study was generated by the Illumina HiSeq Xten (Illumina, Inc., San Diego, CA, USA) platform. |
|---|---|
| Data analysis | All the softwares used in this study are cited in the manuscript and listed as follows: GetOrganelle toolkit version 1.7.3.5., MAFFT program version 7, MEGA X, minimap2 v2.10, sambamba v0.6.8, GATK, version v4.0.12.0, vcftools v 0.1.19, bcftools v 1.10.2, PopLDdecay v3.31, shuf version 8.22, HaploBlocker v1.5.18, ade4 v 1.7-16, PLINK v1.90b6.9, IQ-TREE v 1.6.12, ADMIXTURE v 1.30, Eigenstrat package v 7.2.1, QTL Icimapping v4.2, MapQTL v6.0, Joinmap v5.0, GAPIT v 3.0, R v4.1.3. The custom script used to extract polymorphisms between and across A. hypogaea subspecies Ahh and Ahf is available at the Zenodo repository (https://doi.org/10.5281/zenodo.12614808)105.The in-house generated python scripts used to count SSRs, calculate their percentage, and perform clustering based on SSR data are available at the Zenodo repository (https://doi.org/10.5281/zenodo.12191309)106. |

For manuscripts utilizing custom algorithms or software that are central to the research but not yet described in published literature, software must be made available to editors and reviewers. We strongly encourage code deposition in a community repository (e.g. GitHub). See the Nature Portfolio guidelines for submitting code & software for further information.

## Data

Policy information about <u>availability of data</u>

All manuscripts must include a <u>data availability statement</u>. This statement should provide the following information, where applicable:

- Accession codes, unique identifiers, or web links for publicly available datasets
- A description of any restrictions on data availability
- For clinical datasets or third party data, please ensure that the statement adheres to our <u>policy</u>

> The datasets analyzed or generated by this study are available in Supplementary Data and the public repositories of the National Center for Biotechnology Information (NCBI, https://www.ncbi.nlm.nih.gov) and Zenodo (https://zenodo.org/). Whole genome re-sequencing data are available at the NCBI Sequence Read Archive (SRA) database (Bioproject PRJNA605106). The assembled chloroplast genomes obtained in this study are available at the NCBI GenBank database (accession numbers from PP971404 to PP971516). Genomic SNPs and InDels identified in this study are available at the Zenodo repository (https://doi.org/10.5281/zenodo.12475904).

## Research involving human participants, their data, or biological material

Policy information about studies with <u>human participants or human data</u>. See also policy information about <u>sex, gender (identity/presentation), and sexual orientation</u> and <u>race, ethnicity and racism</u>.

| | |
|---|---|
| Reporting on sex and gender | *Use the terms sex (biological attribute) and gender (shaped by social and cultural circumstances) carefully in order to avoid confusing both terms. Indicate if findings apply to only one sex or gender; describe whether sex and gender were considered in study design; whether sex and/or gender was determined based on self-reporting or assigned and methods used. Provide in the source data disaggregated sex and gender data, where this information has been collected, and if consent has been obtained for sharing of individual-level data; provide overall numbers in this Reporting Summary. Please state if this information has not been collected. Report sex- and gender-based analyses where performed, justify reasons for lack of sex- and gender-based analysis.* |
| Reporting on race, ethnicity, or other socially relevant groupings | *Please specify the socially constructed or socially relevant categorization variable(s) used in your manuscript and explain why they were used. Please note that such variables should not be used as proxies for other socially constructed/relevant variables (for example, race or ethnicity should not be used as a proxy for socioeconomic status). Provide clear definitions of the relevant terms used, how they were provided (by the participants/respondents, the researchers, or third parties), and the method(s) used to classify people into the different categories (e.g. self-report, census or administrative data, social media data, etc.) Please provide details about how you controlled for confounding variables in your analyses.* |
| Population characteristics | *Describe the covariate-relevant population characteristics of the human research participants (e.g. age, genotypic information, past and current diagnosis and treatment categories). If you filled out the behavioural & social sciences study design questions and have nothing to add here, write "See above."* |
| Recruitment | *Describe how participants were recruited. Outline any potential self-selection bias or other biases that may be present and how these are likely to impact results.* |
| Ethics oversight | *Identify the organization(s) that approved the study protocol.* |

Note that full information on the approval of the study protocol must also be provided in the manuscript.

# Field-specific reporting

Please select the one below that is the best fit for your research. If you are not sure, read the appropriate sections before making your selection.

☒ Life sciences  ☐ Behavioural & social sciences  ☐ Ecological, evolutionary & environmental sciences

For a reference copy of the document with all sections, see <u>nature.com/documents/nr-reporting-summary-flat.pdf</u>

# Life sciences study design

All studies must disclose on these points even when the disclosure is negative.

| | |
|---|---|
| Sample size | The germplasm panel used in this study included 34 accessions of wild diploid species, two accessions of wild tetraploid A. monticola, 353 accessions of cultivated tetraploid A. hypogaea, which were collected from the China and 26 other countries . |
| Data exclusions | No data exclusions. Sequencing data was quality filtered, as described in manuscript. |
| Replication | All the experiments were performed using independent biological replicates as indicated in the manuscript, figure and table legends, and supplementary information data. |
| Randomization | A randomized complete block design was used for phenotyping. |

| Blinding | Blinding was not relevant to our study. |

# Reporting for specific materials, systems and methods

We require information from authors about some types of materials, experimental systems and methods used in many studies. Here, indicate whether each material, system or method listed is relevant to your study. If you are not sure if a list item applies to your research, read the appropriate section before selecting a response.

## Materials & experimental systems

| n/a | Involved in the study |
|-----|------------------------|
| ☒ ☐ | Antibodies |
| ☒ ☐ | Eukaryotic cell lines |
| ☒ ☐ | Palaeontology and archaeology |
| ☒ ☐ | Animals and other organisms |
| ☒ ☐ | Clinical data |
| ☒ ☐ | Dual use research of concern |
| ☐ ☒ | Plants |

## Methods

| n/a | Involved in the study |
|-----|------------------------|
| ☒ ☐ | ChIP-seq |
| ☒ ☐ | Flow cytometry |
| ☒ ☐ | MRI-based neuroimaging |

## Dual use research of concern

Policy information about dual use research of concern

### Hazards

Could the accidental, deliberate or reckless misuse of agents or technologies generated in the work, or the application of information presented in the manuscript, pose a threat to:

| No | Yes | |
|----|-----|---|
| ☒ | ☐ | Public health |
| ☒ | ☐ | National security |
| ☒ | ☐ | Crops and/or livestock |
| ☒ | ☐ | Ecosystems |
| ☒ | ☐ | Any other significant area |

### Experiments of concern

Does the work involve any of these experiments of concern:

| No | Yes | |
|----|-----|---|
| ☒ | ☐ | Demonstrate how to render a vaccine ineffective |
| ☒ | ☐ | Confer resistance to therapeutically useful antibiotics or antiviral agents |
| ☒ | ☐ | Enhance the virulence of a pathogen or render a nonpathogen virulent |
| ☒ | ☐ | Increase transmissibility of a pathogen |
| ☒ | ☐ | Alter the host range of a pathogen |
| ☒ | ☐ | Enable evasion of diagnostic/detection modalities |
| ☒ | ☐ | Enable the weaponization of a biological agent or toxin |
| ☒ | ☐ | Any other potentially harmful combination of experiments and agents |

