## [Peer Review File · Nature Genetics]

Peer Review Information

Manuscript Title: Chloroplast and whole-genome sequencing sheds light on peanut evolutionary history and phenotypic diversification

Corresponding author name(s): Dr Zheng Zheng, Dr Xinyou Zhang

Reviewer Comments & Decisions:

Decision Letter, initial version:
--

31st Aug 2022

Dear Dr Zheng,

Your Article, "DNA sequencing sheds light on the evolutionary history of peanut and identifies genes associated with phenotypic diversification" has now been seen by 2 referees. You will see from their comments copied below that while they find your work of considerable potential interest, they have raised quite substantial concerns that must be addressed. In light of these comments, we cannot accept the manuscript for publication, but would be very interested in considering a substantially revised version that fully addresses these serious concerns.

We hope you will find the referees' comments useful as you decide how to proceed. If you wish to submit a substantially revised manuscript, please bear in mind that we will be reluctant to approach the referees again in the absence of major revisions.

To guide the scope of the revisions, the editors discuss the referee reports in detail within the team with a view to identifying key priorities that should be addressed in revision. In this case, we think both referees have provided constructive reviews aimed at strengthening the analyses and improving the presentation. We particularly ask that you improve the validity of the evolutionary analysis and the identification of candidate genes (Reviewer #1), emphasize the novelty of the study (Reviewer #2), and address all referee comments as thoroughly as possible with appropriate revisions. We hope that you will find the prioritized set of referee points to be useful when revising your study.

If you choose to revise your manuscript taking into account all reviewer and editor comments, please highlight all changes in the manuscript text file. At this stage we will need you to upload a copy of the manuscript in MS Word .docx or similar editable format.

We are committed to providing a fair and constructive peer-review process. Do not hesitate to contact us if there are specific requests from the reviewers that you believe are technically impossible or

unlikely to yield a meaningful outcome.

*2) If you have not done so already please begin to revise your manuscript so that it conforms to our Article format instructions, available here. Refer also to any guidelines provided in this letter.

Please be aware of our guidelines on digital image standards.

[redacted]

If you wish to submit a suitably revised manuscript we would hope to receive it within 6 months. If you cannot send it within this time, please let us know. We will be happy to consider your revision so long as nothing similar has been accepted for publication at Nature Genetics or published elsewhere. Should your manuscript be substantially delayed without notifying us in advance and your article is eventually published, the received date would be that of the revised, not the original, version.

Thank you for the opportunity to review your work.

Sincerely,

Wei

Wei Li, PhD
Senior Editor
Nature Genetics
New York, NY 10004, USA
www.nature.com/ng

Reviewers' Comments:

Reviewer #1:

Remarks to the Author:

The manuscript "DNA sequencing sheds light on the evolutionary history of peanut and identifies genes associated with phenotypic diversification" assembled chloroplast genomes of 113 peanut accessions and re-sequenced 353 peanut accessions from different countries. They used the chloroplast genomes as well as the re-sequencing data to study the evolutionary relationship of different peanut subspecies. Moreover, they carried out QTL mapping and GWAS to identify genes corresponding to the phenotypic diversification of peanuts. The function of one of the candidate genes corresponding to the peanut's inner integument color was verified in the heterologous *Arabidopsis* system.

Major comments:

(1) My main concern is whether the evolutionary analysis based on chloroplast genomes can support the inference of peanut evolutionary history? As an important part of the manuscript, the origin and evolution of cultivated peanuts are mainly based on the analysis of the chloroplast genomes.

The first question related to this issue is that for the AABB type (hybrid), what chloroplast genome was retained? For example, if it retained the chloroplast from the A genome donor, how could the authors compare it to the chloroplast of the B genome donor?

In a recent paper on legume genomics (*New Phytologist* 225:1355–1369), it was stated in the first paragraph of their discussions "Clearly, moving beyond the chloroplast genome and analysing nuclear gene data is necessary to improve phylogenetic hypotheses for the legume family, as found for other parts of the plant tree of life where chloroplast data have proven insufficiently informative".

Considering the quantity of chloroplast genome variations and genetic limitations, whether the topology reflected by the chloroplast genomes can truly represent the evolution of the nuclear genomes, and whether using the homologous genes of the nuclear genome will get different results.

In relation to this, my second major question is related to the accuracy of the data. In the phylogenetic tree in Fig. 1b built with chloroplast genomes, the tree exhibits quite some ambiguity. Accessions from *A. duranensis* and *A. archeri* have been placed between *A. hypogaea* var. *hyp* and *A. hypogaea* var. *fas*; the insertion of non-species between different varieties of the same species

suggests that the tree may have errors. Also, there are mixing-ups of *A. hypogaea* var. *hyp* and *A. hypogaea* var. *fas*, and such an abnormality was not observed when whole genome SNPs were used for the phylogenetic studies (Fig. 1c). Furthermore, as a donor of the B genome, *A. ipaensis* seems to be very far away from the genome of cultivated peanuts with the AABB genome.

Therefore, more evidence is needed to justify the use of chloroplast genomes for phylogenetic analysis in the study.

(2) Another major finding of this manuscript is on the QTL analysis of some important agronomic traits using both GWAS and a recombinant inbred population. QTL identification using the RI population (from two parents) does not seem to fit the context of this paper. So we shall focus on the analysis of GWAS.

The authors claimed multiple times in the manuscript that some GWAS results were not correct due to the assembly error involving homoeologous regions. First, the authors should provide evidence to demonstrate the assembly error. For example, the association of chromosome 2 to the flowering pattern is much stronger than that in chromosome 12. However, the authors claimed a misassembly in chromosome 2 and completely ignored the region. Can the markers on the genetic map be mapped to chromosomes 2 and 12 simultaneously? The authors cannot rule out the possibilities that the polymorphisms of the homoeologous genes in the homoeologous regions also contribute to the phenotype. This happens in other crop species.

The candidate interval may have many genes and variations related to the traits. Whether the candidate genes directly contribute to the trait or not, it is challenging to meet the current standard requirements of NG without rigorous experimental verification.

The logic of identification of the final candidate genes in QTLs from GWAS analysis is very unclear. The QTLs are of large intervals and each contains many genes and variations. The authors should provide experimental proofs before making claims on causal candidate genes.

Among all the genes identified to associate with phenotypic diversification, the function of only one gene was tested in the heterologous *Arabidopsis* system. Due to the use of 35S constitutive promoter, incomplete complementation of the *Arabidopsis* mutant phenotype, lack of biological repeat information, and the lack of support with biochemical data, the result only partially supported the function of the gene of interest in the native host. More rigorous experiments should be done.

Page 8 Lines 16-24. The MADS gene potentially associated with growth habit was identified through QTL studies with the function unproved. The data here did not support that the mutations of this gene (2 bp insertion or 1870 bp deletion) were associated with the phenotype of interest. Both Fig. 4 and Supplementary Fig. 7 showed only the polymorphisms, but not the association.

Minor comments:

(3) The figures in both the main text and the supplementary file seem to be hard to load using Acrobat Reader.

(4) The introduction part should include information on the significance of the samples selected, especially for those samples used in constructing the origin and evolution story.

- (5) A total of 1,884 polymorphisms were found between the 113 assembled chloroplast genomes and most of the polymorphic sites occurred between wild and cultivated peanuts. Are these polymorphisms important for shaping the cultivated peanuts?
- (6) The peanut pictures in figure 1c need to be connected with specific accessions.
- (7) Figure 1e, the LD analysis, is the influence of sample number considered? How many samples are there in each group? Has it been standardized?
- (8) Page 5, lines 4-7, data access needs to be moved to a separate data access part.
- (9) The construction process of mapping populations, the genetic markers and specific QTL methods need to be carefully described in the method.
- (10) For all the QTL mapping results using two genetic populations, the genetic length of the chromosome of the two populations differed so much. What were the reasons?
- (11) The current discussion section is too simple. The significance of this study and the implication of the results should be discussed in more detail.
- (12) Methods: A lot of information is missing in the methods section. For example, the information on the biparental populations used in QTL mapping, the methods for the biparental QTL mapping, the construction of transgenic Arabidopsis etc are missing in the methods section. Details of protocols, pipeline, quality check criteria etc should also be given.
- (13) PAGE 12 Line 34. The assembled repeat pattern of chloroplastic genomes of different accessions should be part of the results instead of methods.

Reviewer #2:

Remarks to the Author:

This paper presents an impressive sequencing effort of a worldwide economically central crop. Methods and presentation are clear, and the writing is overall fairly clear. The work will of course be of great interest to peanut breeders and scientists working with peanut, and the sequencing effort of the paper is prodigious. But this alone of course does not give it broad interest, nor do I feel obviously motivates publication in the highest profile journals.

The problem is that I see no clear interest to biologists outside of the peanut breeding community, and indeed I see no clear lessons in either the data or analysis that inform any other system, as presented. If the authors draw out a story that engages non-peanut scientists as to what the real novelty (beyond a massive peanut resource) may be, I would like to be more enthusiastic. As it is, however, this paper is severely limited to providing new resources about peanut and a nice description of peanut demography and candidate alleles underlying certain traits. It tells us little that I can see about fundamental biology that other scientists may take away from reading it.

Of the candidate genes explored, for plant biologists the TFL1 finding is interesting, but also not

surprising. It would stand as the most expected gene to control such a trait like flowering pattern. I note that the later references cited, in 2015 and 2020 seem inappropriate, as this role for TFL1 was well known before 2010.

Overall, this work appears sound, is well presented, and obviously represents a lot of sequencing, I do not see it of broad interest and would think it's much more appropriate to publish it in a top specialty journal.

Reviewer #3:
None

Author Rebuttal to Initial comments

Reviewer #1:

Remarks to the Author:

The manuscript “DNA sequencing sheds light on the evolutionary history of peanut and identifies genes associated with phenotypic diversification” assembled chloroplast genomes of 113 peanut accessions and re-sequenced 353 peanuts accessions from different countries. They used the chloroplast genomes as well as the re-sequencing data to study the evolutionary relationship of different peanut subspecies. Moreover, they carried out QTL mapping and GWAS to identify genes corresponding to the phenotypic diversification of peanuts. The function of one of the candidate genes corresponding to the peanut's inner integument color was verified in the heterologous *Arabidopsis* system.

(1) My main concern is whether the evolutionary analysis based on chloroplast genomes can support the inference of peanut evolutionary history? As an important part of the manuscript, the origin and evolution of cultivated peanuts are mainly based on the analysis of the chloroplast genomes.

The first question related to this issue is that for the AABB type (hybrid), what chloroplast genome was retained? For example, if it retained the chloroplast from the A genome donor, how could the authors compare it to the chloroplast of the B genome donor?

>>> *A. hypogaea* has just one chloroplast genome, so we did not choose a chloroplast genome to retain. Previous studies indicated that *A. duranensis* donated the chloroplast genome to *A. hypogaea*, in addition to the A nuclear genome (e.g. Kochert et al. 1996, American journal of Botany 83:1282:1291; Grabile et al. Plant Syst. Evol., 298: 1151-1165; Bertioli et al. 2019, Nature Genetics 51: 877-884; Tian et al. 2021, Frontiers in Plant Science 12:804568). Our results

support this notion, as *A. duranensis* is the wild species most closely related to *A. hypogaea* in the chloroplast genome phylogenetic tree.

If the two *Arachis hypogaea* subspecies (*Arachis hypogaea* subsp. *hypogaea*, further referred to as *Ahh*, and *Arachis hypogaea* subsp. *fastigiata*, further referred to as *Ahf*) originated from a single polyploidization event, then all the *A. duranensis* accessions would have been expected to diverge before the split between *Ahh* and *Ahf* in the chloroplast phylogenetic tree. In contrast, as a main finding of our work, we found four *A. duranensis* accessions (PI219823, PI468201, PI468202 and PI604844) in the *Ahh* phylogenetic clade (Fig. 1b), indicating that different *A. duranensis* mother plants, and thus different allopolyploidization events, gave origin to *Ahh* and *Ahf*. We revised the “Results” section of the manuscript to clarify this concept.

As there were some wild species showing unexpected clustering in the chloroplast phylogenesis, we carefully checked all the wild species karyotypes based on available literature and our unpublished data (see the revised Supplementary Table 1, and GISH photos and explanations provided in Figure1 of the Figures for response to the reviewer).

In a recent paper on legume genomics (New Phytologist 225:1355–1369), it was stated in the first paragraph of their discussions “Clearly, moving beyond the chloroplast genome and analysing nuclear gene data is necessary to improve phylogenetic hypotheses for the legume family, as found for other parts of the plant tree of life where chloroplast data have proven insufficiently informative”.

>>> Chloroplast genomes are maternally inherited, therefore chloroplast genome phylogenesis is more desirable than nuclear phylogenesis to address specific research questions such as maternal evolutionary dynamics and allopolyploidization, as previously stated (Tian et al. 2021 Front. Plant Sci. 12: 804568; Moore et al. 2007, Proc. Natl. Acad. Sci U.S.A. 104:19363). In the revised version of the manuscript, we clearly state that we used chloroplast phylogenesis to investigate a specific research question, i.e. whether the same mother plant contributed to the allopolyploidization of *Ahh* and *Ahf*.

Considering the quantity of chloroplast genome variations and genetic limitations, whether the topology reflected by the chloroplast genomes can truly represent the evolution of the nuclear genomes, and whether using the homologous genes of the nuclear genome will get different results.

>>> The relatively limited number of SNP variation at the basis of chloroplast phylogenesis is expected, as the chloroplast genome is highly conserved among land plants. To exclude the possibility that sequencing errors affected the topology of the chloroplast tree, in the revised version of the manuscript we report the results of further validation of chloroplast SNP

polymorphisms in wild and cultivated peanut by KASP technology (see the new Supplementary Table 5 and Supplementary Fig. 1).

As suggested by the referee, the revised version of the manuscript integrates chloroplast genome phylogenesis with nuclear genome phylogenesis (see the new Supplementary Figure S3). Two different nuclear phylogenetic trees were constructed: the first using chromosomes 1-10 of the tetraploid *A. hypogaea* (corresponding to the A genome) and the diploid A genome of *A. duranensis*; the second using chromosomes 11-20 of *A. hypogaea* (corresponding to the B genome) and *A. ipaensis*, the wild species that is known to be the donor of the B genome (e.g. Bertioli et al. 2016, Nat. Genet. 48:438-446). In accordance with the chloroplast genome phylogenesis, both nuclear genome phylogenesis clearly distinguish *Ahh* from *Ahf*. However, differently from the chloroplast tree, the nuclear tree based on the A genome does not include *A. duranensis* accessions within the *Ahh* clade. In the revised version of the manuscript, we discuss that this difference is possibly due to recombination between *Ahh* and *Ahf* and rearrangements between the A and B genome after the polyploidization. Importantly, in support of this hypothesis, Bertioli et al. (2016, Nat. Genet. 48:438-446) and Bertioli et al. (2019, Nat. Genet. 51:877-884) already showed the occurrence of homeologous recombination in *A. hypogaea*, which changed the genomic formula of specific chromosomal regions from the expected AABB to AAAA or BBBB.

We finally would like to highlight that there are several examples in literature of incongruencies between chloroplast and nuclear phylogenies (e.g. Hodel et al. 2022, Front. Plant Sci. 12:820997), as they result from different evolutionary forces.

To provide different evidence supporting the results of chloroplast phylogenesis, indicating distinct allopolyploidization events at the basis of the evolution of the peanut subspecies *Ahh* and *Ahf*, we investigated the distribution of genomic polymorphisms (see the new Figures 1e-j and Supplementary Figure 4). Bootstrap sampling of groups of individuals from *Ahh* and *Ahf* allowed to reveal a large excess of polymorphisms between groups (P_B) compared with polymorphisms shared across groups (P_A), in accordance with a scenario in which alleles that were polymorphic between different pairs of tetraploid progenitors were fixed in *Ahh* and *Ahf* (Fig. 1e and 1f). In contrast, sampling of group pairs within the same subspecies yielded opposite results ($P_B \ll P_A$) (Fig. 1g-j), in accordance with their descendance from a common tetraploid progenitor.

In relation to this, my second major question is related to the accuracy of the data. In the phylogenetic tree in Fig. 1b built with chloroplast genomes, the tree exhibits quite some ambiguity. Accessions from *A. duranensis* and *A. archeri* have been placed between *A. hypogaea* var. *hyp* and *A. hypogaea* var. *fas*; the insertion of non-species between different varieties of the same species suggests that the tree may have errors. Also, there are mixing-ups of *A. hypogaea*

var. hyp and *A. hypogaea* var. fas, and such an abnormality was not observed when whole genome SNPs were used for the phylogenetic studies (Fig. 1c). Furthermore, as a donor of the B genome, *A. ipaensis* seems to be very far away from the genome of cultivated peanuts with the AABB genome.

>>> As stated above, we performed further KASP assays that confirmed the true polymorphisms between the wild and cultivated peanuts (see Supplementary Table S5). We also retrieved data from previous studies, assembling the chloroplast genome of some of the accessions also considered in our works with a different software. After aligning the sequences, we confirmed the same polymorphic sites. Thus, we are confident that data used for chloroplast genome phylogenesis are reliable.

As said above, the clustering of wild species within the *Ahh* clade is a remarkable finding indicating that *Ahh* arose from a polyploidization event different from the one originating *Ahf*.

As for a few individuals of *Ahh* clustering in the *Ahf* clade and vice versa, we point out that crossing between the subspecies may lead to a situation in which the genome and the plant phenotype is (mostly) referable to one subspecies, whereas the chloroplast genome refers to the other subspecies. Indeed, by pedigree checking we found that cross between *Ahh* and *Ahf* was at the basis of the origin of the accession N524, which inherited the chloroplast genome from the accession N524 (see Supplementary Figure S2).

As for the genetic divergence of *A. ipaensis* in the chloroplast genome tree, this is expected as *A. duranensis*, and not *A. ipaensis*, is the donor of the peanut chloroplast genome.

Therefore, more evidence is needed to justify the use of chloroplast genomes for phylogenetic analysis in the study.

>>> As stated above, we believe that chloroplast genome phylogenesis is more appropriate than nuclear genome phylogenesis to reconstruct the maternal origin of tetraploid cultivated peanut. In addition we provided evidence, after Bertioli et al. 2019 (Nat. Genet. 51:877-884), indicating that homeologous recombination between the peanut sub-genomes can have an impact on the topology of nuclear genome phylogenetic trees. Dissecting genetic polymorphisms between and within the two peanut subspecies *Ahh* and *Ahf* confirmed the results of chloroplast phylogenetic analysis, indicating that different polyploidization events originated *Ahh* and *Ahf* (see the new Figures 1e-j).

(2) Another major finding of this manuscript is on the QTL analysis of some important agronomic traits using both GWAS and a recombinant inbred population. QTL identification using the RI population (from two parents) does not seem to fit the context of this paper. So we shall focus on the analysis of GWAS.

>>> We understand this point of criticism. The choice to integrate GWAS with RIL mapping comes from the need to increase the confidence of GWAS results, as we experienced that GWAS in peanut is more complicated than in other crops. Indeed, the A and B genomes are highly similar (93.11% according to Bertoli et al. 2016, Nat. Genet. 48:438-446); in addition homeologous recombination can change the genomic formula from the expected AABB to AAAA or BBBB. All of this might lead to GWAS signals on wrong chromosomes. In contrast, the linkage approach, although providing a lower mapping resolution, provides more robust chromosomal associations.

The authors claimed multiple times in the manuscript that some GWAS results were not correct due to the assembly error involving homoeologous regions. First, the authors should provide evidence to demonstrate the assembly error. For example, the association of chromosome 2 to the flowering pattern is much stronger than that in chromosome 12. However, the authors claimed a misassembly in chromosome 2 and completely ignored the region. Can the markers on the genetic map be mapped to chromosomes 2 and 12 simultaneously? The authors cannot rule out the possibilities that the polymorphisms of the homoeologous genes in the homoeologous regions also contribute to the phenotype. This happens in other crop species.

>>> We carefully addressed this point of criticism. In the revised version of the manuscript, we performed QTL mapping using an additional RIL population. This resulted in the identification of strong signals on both chromosomes 2 and 12 (see the new Figure 2, panels d and e), whereas the RIL population on considered in the previous version of the manuscript only yielded a signal on chromosome 12 (Fig. 2c). We conclude that the reviewer was right and indeed the GWAS signals on chromosomes 2 and 12 should be both considered, and so the two *TFL1* genes. We acknowledge the reviewer for her/his useful remark.

The candidate interval may have many genes and variations related to the traits. Whether the candidate genes directly contribute to the trait or not, it is challenging to meet the current standard requirements of NG without rigorous experimental verification.

The logic of identification of the final candidate genes in QTLs from GWAS analysis is very unclear. The QTLs are of large intervals and each contains many genes and variations. The authors should provide experimental proofs before making claims on causal candidate genes.

>>> Unfortunately, peanut is recalcitrant to regeneration and genetic transformation (Mallikarjuna et al. 2016, Plant Cell, Tissue and Organ Culture (PCTOC), 125:399–416), and this makes the functional characterization of candidate genes a hard task. Nonetheless, we understood and seriously addressed this point of criticism, and took actions in accordance.

As for the inner tegument color trait, we developed three additional (independent) transgenic *Arabidopsis* lines using the candidate gene *AhLAC*, and monitored transgene expression by real-time qPCR. As shown in the new Supplementary Fig. S9, all the transgenic lines complemented the *Arabidopsis* mutant (although partially). In addition, the tegument colour roughly correlated with the transgene expression level. Finally, a KASP assay was designed on the MITE insertion (Supplementary Tables 18) and verified to be fully co-segregating with the inner tegument color in both the GWAS population and the YZ9102×wt09-0023 RIL population (Supplementary Tables 19-21).

As for the other candidate genes for the flowering pattern and growth habit, unfortunately we could not perform heterologous assays in *Arabidopsis*, as we could not find *Arabidopsis* phenotypes related to these traits. We therefore modified the discussion and clearly states that the *TFL1* and *MADS-box* genes found in QTL intervals are putatively controlling flowering pattern and growth habit, respectively. However, to provide further indication that *TFL1* controls the flowering pattern, we performed mining and found full co-segregation of any of these mutation and the sequential pattern (see the new Supplementary Tables 13 and 14).

Finally, we would like to point out that, although there is no conclusive evidence, *TFL1* is an obvious candidate to control the flowering pattern in peanut, as *TFL*-like genes were shown to change meristem indeterminacy across plant species, including legumes (Severin et al. 2010; Dhanasekar et al. 2015, *Molecular Genetics and Genomics* 290:55–65; Krylova et al. 2020, *Bio. Comm.* 66: 85–108).

Among all the genes identified to associate with phenotypic diversification, the function of only one gene was tested in the heterologous *Arabidopsis* system. Due to the use of 35S constitutive promoter, incomplete complementation of the *Arabidopsis* mutant phenotype, lack of biological repeat information, and the lack of support with biochemical data, the result only partially supported the function of the gene of interest in the native host. More rigorous experiments should be done.

>>> Further experimental activities were performed to address this issue. As mentioned above, in the revised manuscript we report the occurrence of co-segregation between the inner tegument color and the allelic state of *AhLAC*, in both GWAS and the RIL populations. Moreover, as shown in the new Supplementary Figure S9, we developed three additional 35S::*AhLAC* transgenic *Arabidopsis* lines, all displaying complementation of the *Arabidopsis* mutant phenotype. Finally, as also shown in the new Supplementary Figure S9, we found positive correlation between *AhLAC* expression in the transgenic lines and the tegument color. Overall, we believe that the body of evidence reported in the revised version of the manuscript is sufficient to show that *AhLAC* controls the inner tegument color in peanut.

Page 8 Lines 16-24. The MADS gene potentially associated with growth habit was identified through QTL studies with the function unproved. The data here did not support that the mutations of this gene (2 bp insertion or 1870 bp deletion) were associated with the phenotype of interest. Both Fig. 4 and Supplementary Fig. 7 showed only the polymorphisms, but not the association.

>>> We agree with this point of criticism. In the revised version of the manuscript, we made clear that the *MADS-box* gene is just a candidate for controlling the growth habit, and further investigations should be carried out to prove its function.

Minor comments:

(3) The figures in both the main text and the supplementary file seem to be hard to load using Acrobat Reader.

>>> We reduced the figure size in the PDF files by compressed the pictures.

(4) The introduction part should include information on the significance of the samples selected, especially for those samples used in constructing the origin and evolution story.

>>> We modified the introduction part accordingly. We now specify that the collection was selected to encompass several diploid *Arachis* species, the wild tetraploid species *A. monticola*, and *A. hypogaea*. Further information on the accessions selected to perform analyses are specified in the text.

(5) A total of 1,884 polymorphisms were found between the 113 assembled chloroplast genomes and most of the polymorphic sites occurred between wild and cultivated peanuts. Are these polymorphisms important for shaping the cultivated peanuts?

>>> Genes associated with the peanut phenotypic traits considered in our study, important for diversification between the two peanut sub-species and breeding, are nuclear, as they were identified through GWAS and linkage mapping. We cannot exclude that some chloroplast polymorphism is also controlling phenotypic traits; in our study, chloroplast polymorphism information was only used for phylogenesis.

(6) The peanut pictures in figure 1c need to be connected with specific accessions.

>>> We have changed the figures according to this suggestion, we acknowledge the reviewer for this remark.

(7) Figure 1e, the LD analysis, is the influence of sample number considered? How many samples are there in each group? Has it been standardized?

>>> We acknowledge the reviewer for this remark. In the revised manuscript, we added details about LD analysis in the M%M section, as: “LD decay was calculated for all pairs of variations on var. *hypogaea* and irregular-*hypogaea*-type (183 samples), var. *hirsuta* (12 samples), var. *fastigiata* (26 samples), var. *vulgaris* and irregular-*fastigiata*-type (130 samples). Considering the influence of the different number of samples in LD decay calculation, we standardized the sample size of var. *hypogaea* and irregular-*hypogaea*-type and var. *vulgaris* and irregular-*fastigiata*-type to 12 and 26 respectively using shuf (version 8.22) and repeated 100 times.

(8) Page 5, lines 4-7, data access needs to be moved to a separate data access part.

>>> We have now moved this part to the Data availability Section. We acknowledge the reviewer for this suggestion.

(9) The construction process of mapping populations, the genetic markers and specific QTL methods need to be carefully described in the method.

>>> We have added the process and methods used in the method part with the suggestion of reviewer. Thanks for the suggestion!

(10) For all the QTL mapping results using two genetic populations, the genetic length of the chromosome of the two populations differed so much. What were the reasons?

>>> The two RIL populations originated by crossing different parental lines (YZ9102 and YH15 are irregular-*fastigiata*-type and irregular-*hypogaea*-type, respectively, whereas wt09-0023 and W1202 belong to subsp. *hypogaea*.var. *hypogaea*). Then, it is likely that the rate and distribution of the recombination events leading to the two RIL populations are different, leading to different genetic distances expressed in centi Morgan.

(11) The current discussion section is too simple. The significance of this study and the implication of the results should be discussed in more detail.

>>> The discussion was revised accordingly.

(12) Methods: A lot of information is missing in the methods section. For example, the information on the biparental populations used in QTL mapping, the methods for the biparental QTL mapping, the construction of transgenic *Arabidopsis* etc are missing in the methods section. Details of protocols, pipeline, quality check criteria etc should also be given.

>>> We revised the methods section according to this useful suggestion.

(13) PAGE 12 Line 34. The assembled repeat pattern of chloroplastic genomes of different accessions should be part of the results instead of methods.

>>> We removed this part from the revised version of the manuscript, as the structure of peanut chloroplasts was already known (Yin et al. 2017, Sci. Rep. 7:11649).

Reviewer #2:

Remarks to the Author:

This paper presents an impressive sequencing effort of a worldwide economically central crop. Methods and presentation are clear, and the writing is overall fairly clear. The work will of course be of great interest to peanut breeders and scientists working with peanut, and the sequencing effort of the paper is prodigious. But this alone of course does not give it broad interest, nor do I feel obviously motivates publication in the highest profile journals.

The problem is that I see no clear interest to biologists outside of the peanut breeding community, and indeed I see no clear lessons in either the data or analysis that inform any other system, as presented. If the authors draw out a story that engages non-peanut scientists as to what the real novelty (beyond a massive peanut resource) may be, I would like to be more enthusiastic. As it is, however, this paper is severely limited to providing new resources about peanut and a nice description of peanut demography and candidate alleles underlying certain traits. It tells us little that I can see about fundamental biology that other scientists may take away from reading it.

Of the candidate genes explored, for plant biologists the TFL1 finding is interesting, but also not surprising. It would stand as the most expected gene to control such a trait like flowering pattern.

>>> We understand and seriously addressed this point of criticism. As mentioned by the reviewer, our work is expected to be of major interest for the scientists and breeders working on peanut, as it sheds light on the peanut evolutionary history and phenotypic diversification. Importantly, our finding that *Ahh* and *Ahf* originate from different polyploidization events provides an explanation to the contradictory findings from Bertoli et al. (2016) and Zhuang et al. (2019), tracing back peanut polyploidization <10,000 years ago and 0.42-0.47 million years ago, respectively, which were the object of commentary papers previously published in Nature Genetics (see the “Matters Arising” letters from Bertoli et al. 2020, Nature Genetics 52:557-559, and the reply from Zhuang et al. 2020, Nature Genetics 52:560-563). Indeed, the two research

groups based their evolutionary analyses on different reference genome sequences, from the *Ahh* cultivar Tifrunner and the *Ahf* cultivar Shitouqi. This aspect was remarked in the revised manuscript discussion.

The revised version of the manuscript was also amended to point out aspects that might be of interest outside the peanut scientific community. Firstly, we believe that the methodologies used in our work to infer peanut origin from wild progenitors might inspire further research on allopolyploid species whose origin is still elusive or debated. Indeed, chloroplast genome phylogenesis was carried out with several accessions of the chloroplast diploid progenitor, and this was determinant to infer different polyploidization events at the basis of sub-specific taxonomic groups. The revised version of the manuscript provides further evidence indicating different polyploidization of *Ahh* and *Ahf*, based on the study of polymorphisms across and within subspecies (see new Fig. 1e-j). To the best of our knowledge, such an approach was never used to infer the evolutionary history of allopolyploid crops.

As for the gene functions elucidated by our study, we agree with the referee that *TFL1* was an obvious candidate to control the flowering pattern in peanut, although the peanut raceme inflorescence displays distinctive features. However, our work also provides evidence that a laccase gene (named *AhLAC*) controls the inner tegument colour, likely by affecting phenolic composition through the oxidative polymerization of flavonoids (Pourcel et al. 2005, *Plant Cell* 17: 2966–2980), and this can have several physiologic and economic implications. For example, the tegument colour was previously associated with important traits in legumes, such as seed dormancy, response to pathogens, and grain nutritional properties (Smykal et al. 2014, *Frontiers in Plant Science* 5:351). Thus, we believe that this finding might be of broad interest for scientists and breeders outside the peanut community.

We point out that the revised version of the manuscript also provides further evidence for the *AhLAC* function. Indeed, we developed three additional (independent) transgenic *Arabidopsis* lines using *AhLAC*, and monitored transgene expression by real-time qPCR. As shown in the new Supplementary Figure S9, all the transgenic lines complemented the *Arabidopsis* mutant. In addition, the tegument colour roughly correlated with the transgene expression level. Finally, we performed allele mining of the two *AhLAC* mutations in both the GWAS and the YZ9102×wt09-0023 RIL populations by KASP assays, and found full co-segregation between the presence of at least one mutation and the transparent tegument (see the new Supplementary Table 19-21).

I note that the later references cited, in 2015 and 2020 seem inappropriate, as this role for *TFL1* was well known before 2010.

>>> Thanks for the suggestion, we have added the reference “Shannon and Meeks-Wagner, 1991, *Plant Cell* 3:877-892” in the manuscript.

Overall, this work appears sound, is well presented, and obviously represents a lot of sequencing, I do not see it of broad interest and would think it's much more appropriate to publish it in a top specialty journal.

>>> Thank you for appreciating our work. We hope that the revised version of the manuscript, also presenting changes suggested by the other reviewer, is suitable for publication.

Reference

- Kochert, G. et al. RFLP and cytogenetic evidence on the origin and evolution of allotetraploid domesticated peanut, *Arachis hypogaea* (Leguminosae). *Am. J. Bot.* **83**, 1282–1291 (1996).
- Grabiele, M., Chalup, L., Robledo, G., Seijo, G. Genetic and geographic origin of domesticated peanut as evidenced by 5S rDNA and chloroplast DNA sequences. *Plant Syst. Evol.* **298**, 1151–1165 (2012).
- Bertioli, D. J. et al. The genome sequence of segmental allotetraploid peanut *Arachis hypogaea*. *Nat. Genet.* **51**, 877–884 (2019).
- Tian, X. et al. Chloroplast Phylogenomic analyses reveal a maternal hybridization event leading to the formation of cultivated peanuts. *Front. Plant. Sci.* **12**, 804568 (2021).
- Moore, M. J., Bell, C. D., Soltis, P. S., Soltis, D. E. Using plastid genome-scale data to resolve enigmatic relationships among basal angiosperms. *Proc. Natl. Acad. Sci. U. S. A.* **104**, 19363 (2007).
- Bertioli, D. J. et al. The genome sequences of *Arachis duranensis* and *Arachis ipaensis*, the diploid ancestors of cultivated peanut. *Nat. Genet.* **48**, 438–446 (2016).
- Hodel, R. G. J., Zimmer, E. A., Liu, B. B., Wen, J. Synthesis of nuclear and chloroplast data combined with network analyses supports the polyploid origin of the apple tribe and the hybrid origin of the maleae-gillenieae clade. *Front Plant Sci.* **12**, 820997 (2022).
- Mallikarjuna, G., Rao, T. S. R. B., Kirti, P.B. Genetic engineering for peanut improvement: current status and prospects. *Plant Cell Tiss. Organ. Cult.* **125**, 399–416 (2016).
- Severin, A. J. et al. RNA-Seq Atlas of *Glycine max*: a guide to the soybean transcriptome. *BMC Plant Biol.* **10**, 160 (2010).
- Dhanasekar, P., Reddy, K. S. A novel mutation in TFL1 homolog affecting determinacy in cowpea (*Vigna unguiculata*). *Mol. Genet. Genomics* **290**, 55–65 (2015).

- Krylova, E. A., Khlestkina, E. K., Burlyaeva, M. O., Vishnyakova, M. A. Determinate growth habit of grain legumes: role in domestication and selection, genetic control. *Ecological genetics* **18**, 43-58 (2020).
- Yin, D., Wang, Y., Zhang, X., Ma, X., He, X., Zhang, J. Development of chloroplast genome resources for peanut (*Arachis hypogaea* L.) and other species of *Arachis*. *Sci Rep.* **7**, 11649 (2017).
- Zhuang, W. et al. The genome of cultivated peanut provides insight into legume karyotypes, polyploid evolution and crop domestication. *Nat Genet* **51**, 865-876 (2019).
- Bertioli, D. J. et al. Evaluating two different models of peanut's origin. *Nat. Genet.* **52**, 557–559 (2020).
- Zhuang, W. et al. Reply to: Evaluating two different models of peanut's origin. *Nat. Genet.* **52**, 560–563 (2020)
- Pourcel, L. et al. TRANSPARENT TESTA10 encodes a laccase-like enzyme involved in oxidative polymerization of flavonoids in *Arabidopsis* seed coat. *Plant Cell* **17**, 2966-2980 (2005).
- Smýkal, P., Vernoud, V., Blair, M. W., Soukup, A., Thompson, R. D. The role of the testa during development and in establishment of dormancy of the legume seed. *Front Plant Sci* **5**, 351 (2014).
- Shannon, S. & Meeks-Wagner, D. R. A. Mutation in the *Arabidopsis* TFL1 gene affects inflorescence meristem development. *Plant Cell* **3**, 877-892 (1991)

Figures for response to reviewers:

Figure 1 Oligonucleotide dye designed to distinguish chromosome A08 karyotypes and results of five wild *Arachis*. (A) For *A. duranensis* as A genome group, there are specific Oligonucleotide signal (pink signal) designed to distinguish chromosome A08 (taking PI 468323 as example); (B) For *A. ipaensis* (taking PI 468322 as example) as B genome group, signals occurred in two pairs of chromosomes with lower density; (C) For *A. stenophylla* with EE genome group, signals occurred in several pairs of chromosomes (taking PI 468178 as example); (D) *A. glabrata* was originally deemed as a homotetraploidy species of the Section *Rhizome* with genome of $R_1R_1R_2R_2$, however, the identification result showed that it contained the "A08 chromosome" and should be a diploid with A genome group for PI 468336; (E) *A. paraguariensis* was originally deemed as a diploid from Section *Erectoides* with the genome of EE group, but PI 262842 contained the "A08 chromosome" and so was identified as a diploid from A genome group.

Decision Letter, first revision:

19th Jun 2023

Dear Dr Zheng,

Your Article, "DNA sequencing sheds light on peanut evolutionary history and phenotypic diversification" has now been seen by 2 referees. You will see from their comments below that while they find your work of interest, some important points are raised by Reviewer #1. We are interested in the possibility of publishing your study in Nature Genetics, but would like to consider your response to these concerns in the form of a revised manuscript before we make a final decision on publication.

We therefore invite you to revise your manuscript taking into account all reviewer and editor comments. Please highlight all changes in the manuscript text file. At this stage we will need you to upload a copy of the manuscript in MS Word .docx or similar editable format.

*2) If you have not done so already please begin to revise your manuscript so that it conforms to our Article format instructions, available

here.

*3) Include a revised version of any required Reporting Summary:

Please be aware of our guidelines on digital image standards.

[redacted]

We hope to receive your revised manuscript within 2 to 3 months. If you cannot send it within this time, please let us know.

Please do not hesitate to contact me if you have any questions or would like to discuss these revisions

further.

Nature Genetics is committed to improving transparency in authorship. As part of our efforts in this direction, we are now requesting that all authors identified as 'corresponding author' on published papers create and link their Open Researcher and Contributor Identifier (ORCID) with their account on the Manuscript Tracking System (MTS), prior to acceptance. ORCID helps the scientific community achieve unambiguous attribution of all scholarly contributions. You can create and link your ORCID from the home page of the MTS by clicking on 'Modify my Springer Nature account'. For more information please visit please visit www.springernature.com/orcid.

Sincerely,
Wei

Wei Li, PhD
Senior Editor
Nature Genetics
New York, NY 10004, USA
www.nature.com/ng

Reviewers' Comments:

Reviewer #1:

Remarks to the Author:

In the revised manuscript, the author used KASP assays to verify the accuracy of chloroplast markers and added a section on resequencing evolutionary analysis. However, the concerns I am focusing on have not been adequately addressed. The conclusion inferred from the evolutionary tree is not convincing because the experimental design for studying evolution in chloroplast genomes, which includes both inter-species samples and a large number of intra-species samples, is not proper. It is more suitable to use chloroplast genomes to study evolutionary relationships between species, but the resolution issues can lead to misjudgment when evolutionary relationships within species are being investigated. Other major concerns are as follows:

1. According to the results, 1884 variations mainly occur between wild and cultivated peanuts, and the Ah genome only has 14 markers. Such a limited number of markers make the evolutionary tree less reliable. However, different from the chloroplast tree, the nuclear tree based on the A genome does not include *A. duranensis* accessions within the Ahh clade. Therefore, the conclusion of the manuscript based on the trees constructed using unsaturated chloroplast markers may not be reliable.
2. When aligning WGS to the genome, alignment performance between Ahh and Ahf needs to be compared to clarify that there are no significant biases.

3. In addition, the subsequent process of gene screening is still unclear. Following up on the candidate intervals obtained from GWAS and QTL, do the intervals contain only the candidate genes described in the manuscript? The selection criteria of functional genes in the manuscript are still unclear.

4. Regarding the scientific issues about whether the same mother plant contributed to the allopolyploidization of Ahh and Ahf, can a clear conclusion be drawn based on the existing results? Have the authors identified the mother plant?

5. Regarding the claim "wild species within the Ahh clade is a remarkable finding indicating that Ahh arose from a polyploidization event different from the one originating Ahf", the author did not provide a proper response to the reviewer's question. Chloroplast genomes cannot be used to distinguish closely related species and it is not suitable for evolutionary research within species. Using KASP assays to verify the chloroplast markers does not provide an answer to this question.

6. A discussion on putative genes (e.g. TFL1) without experimental proof is not convincing. The authors should explore the relevant phenotype (prostrate growth) in other species with a transformation system available.

7. The data in the revised version still involves the use of 35S constitutive promoter for the incomplete complementation in Arabidopsis, and there is no solid evidence to show that AhLAC controls the inner tegument color in peanuts. The authors should use a native promoter to generate transgenic lines and perform a biochemical assay to support the changes in the inner tegument color mediated by AhLAC.

Reviewer #2:

Remarks to the Author:

The revised manuscript by Dr. Zheng and colleagues is a vast improvement on the initial submission. The authors clearly took on board my comments to make much more clear the broad impact of their work, and in this, it is clear that impact goes well outside crop breeders to those working on polyploids generally. They also make clear the interest in the peanut community, addressing an ongoing controversy in the field. Their data and methodology are to my assessment sound and their conclusions are perfectly reasonably worded, representing their findings well. The writing is very clear and engaging. Overall I have no comments for improvement and believe the revised version would be of broad interest.

Author Rebuttal, first revision:

Reviewer #1:

Remarks to the Author:

In the revised manuscript, the author used KASP assays to verify the accuracy of chloroplast markers and added a section on resequencing evolutionary analysis. However, the concerns I am focusing on have not been adequately addressed.

>>>We would like to acknowledge Reviewer #1 for the time and effort spent in reviewing our manuscript. Below, we report point-to-point answers to the issues raised with the revision, and actions we performed to improve the manuscript.

The conclusion inferred from the evolutionary tree is not convincing because the experimental design for studying evolution in chloroplast genomes, which includes both inter-species samples and a large number of intra-species samples, is not proper. It is more suitable to use chloroplast genomes to study evolutionary relationships between species, but the resolution issues can lead to misjudgement when evolutionary relationships within species are being investigated.

>>>We agree with the reviewer that chloroplast genomes are more suitable than nuclear genomes to study inter-specific evolutionary relationships. This actually does not contrast with our choice to use chloroplast genomes for the analysis resulting in Fig. 1b, whose main aim was to investigate inter-specific relationships between the tetraploid *A. hypogaea* and several wild diploid *Arachis* species, thus allowing to reconstruct, from the maternal side, the history of peanut polyploidization.

As the reviewer rightly points out, Fig.1b also reports some intra-specific relationships. In particular, grouping of the two *A. hypogaea* sub-species *Ahh* and *Ahf* in different phylogenetic clades (together with the grouping of several accessions of *A. duranensis* within the *Ahh* clade) was fundamental for our conclusion that different allopolyploidization events originated *Ahh* and *Ahf*. However, we would like to highlight that, in this respect, the results of the chloroplast phylogeny are fully matching with those of the nuclear phylogeny reported in Fig. 1c (both indicate, with maximum bootstrap support, the separation of *Ahh* from *Ahf*).

We seriously considered the issue raised by the reviewer and carefully examined relevant literature. Several authors agree with our choice to use chloroplast genomes to reconstruct the maternal origin of allopolyploid species. We quote here extracts from some articles:

1. Tian et al. (2021) – “Plastomics provide a powerful tool in phylogenetic studies involving particular evolutionary events, such as interspecific hybridization, allopolyploidization, rapid evolution, etc. In contrast to nuclear genomes, plastomes are maternally inherited (Moore et al., 2007). The evolutionary rate of plastomes is low, and there is no recombination during chloroplast division (Daniell et al., 2016). Therefore, plastomes are good resources for studying maternal evolutionary dynamics (Tonti-Filippini et al., 2017)”.
2. Brock et al. (2022) – “Chloroplast genomes of most flowering plants are maternally inherited, making them particularly useful for studying historical plant hybridization and inferring maternal lineages involved in polyploidization events”.
3. Liu and Musial (2001)– “Organelle inheritance is strictly maternal for most plant species. This property makes organelle DNAs ideal material for identifying the maternal parents of polyploid species”.
4. Chen et al. (2021) - “to better understand the maternal contribution to the evolution of the species of *Kengyilia*, it is essential to conduct a good comparative study of chloroplast genome-wide in *Kengyilia* and its relatives, covering nearly all of the genomic combinations in *Triticeae*”.
5. Chen et al. (2020) – “Chloroplast (cp) DNA is usually maternally inherited in angiosperms and has been widely used to identify the maternal parent and mode of hybrid speciation in polyploids by phylogenetic comparison”.

As for cultivated peanut, notably Tian et al. (2021) report inconsistencies among nuclear peanut phylogenies, and thus suggested that “...a study carried out with a different type of sequence data (i.e., plastomic data) ... would be appropriate when trying to reconstruct the phylogeny of cultivated peanuts”. To the best of our capacity, we could not find literature supporting the use of nuclear genome phylogenies to reconstruct the maternal origin of an allopolyploid species.

Literature also provided us several examples of studies using our experimental design to infer the maternal origin of polyploids and multiple polyploidization events. For example, Brock et al. (2022), aiming to determine the diploid maternal contributors of polyploid *Camelina* lineages, carried out a phylogenetic study using 82 chloroplast genome sequences from 7 species. This allowed to reveal the maternal diploid parental species of three allopolyploid species. In addition, the inclusion in the analysis of several accessions of the diploid species *C. hispida* allowed to infer multiple independent hybridization and polyploidization events at the origin of the allotetraploid *C. rumelica*.

We were probably not clear in stating that the main purpose of the chloroplast phylogeny was to reconstruct inter-specific relationships between wild *Arachis* diploid species and the allotetraploid *A. hypogaea*, and this probably generated a misunderstanding. In the revised version of the manuscript, we clearly state that the main purpose of that chloroplast phylogenesis was the reconstruction of peanut maternal origin and allopolyploidization. In addition, we cite several articles supporting the use of chloroplast genomes to infer the maternal origin of allopolyploids.

Finally, we also cite literature indicating that several polyploid species are of multiple origins (e.g. Wolfe et al. 2023; Mavrodiev et al. 2015; Soltis and Soltis 1999), in accordance with our findings.

Besides, as previously stated, we would like to point out that, compared to the original submission, the revised version of the manuscript integrated an additional analysis, independent from chloroplast phylogenesis, also indicating distinct allopolyploidization events at the basis of the evolution *Ahh* and *Ahf* (see Figures 1e-j). Specifically, we performed bootstrap sampling of two groups of individuals (one from *Ahh* and the other from *Ahf*), and detected a large excess of polymorphisms between groups (P_B) compared with polymorphisms shared across groups (P_A), in accordance with a scenario in which alleles that were polymorphic between different pairs of diploid progenitors were fixed in either *Ahh* or *Ahf* (Fig. 1e and 1f). In contrast, bootstrap sampling of group pairs within the same subspecies yielded opposite results ($P_B \ll P_A$) (Fig. 1g-j), in accordance with their descendance from a common tetraploid progenitor.

Other major concerns are as follows:

1. According to the results, 1884 variations mainly occur between wild and cultivated peanuts, and the Ah genome only has 14 markers. Such a limited number of markers make the evolutionary tree less reliable.

>>> We understand this point of criticism, although we would like to point out that the main feature of the chloroplast tree topology supporting our conclusion of independent polyploidization events (i.e. the separation between a clade containing *Ahh/A.duranensis* and a clade containing *Ahf*) was extremely significant at the statistical level (maximum bootstrap support), even if based on 14 markers. To provide more support to our results, in the revised version of the manuscript we extended the analysis of chloroplast variation using mononucleotide repeat (MNR) loci, which are the most abundant class of SSRs in chloroplast genomes. This kind of analysis was already reported by several publications addressing the study of chloroplast genetic variation (e.g. Mao et al. 2023; Zhang et al. 2023; Lian et al. 2022; Mu et al. 2023). We obtained data from more than 10,500 MNR loci ranging from 3 to 18 repeats (new Supplementary Table S6. The results of clustering analysis with these new data (see the new Figure S3 in the revised manuscript) confirmed the separation of *Ahh* from *Ahf*, and the presence of *A. duranensis* in the *Ahh* phylogenetic clade.

However, different from the chloroplast tree, the nuclear tree based on the A genome does not include *A. duranensis* accessions within the *Ahh* clade. Therefore, the conclusion of the manuscript based on the trees constructed using unsaturated chloroplast markers may not be reliable.

>>> As discussed above, the topology of our chloroplast tree was statistically solid and supported by additional chloroplast markers identified and used in the revised version of the manuscript. Moreover, as discussed above, literature encourages the use of chloroplast, rather than nuclear, phylogenesis, to reconstruct the maternal origin of polyploids. There might be several reasons explaining different topologies between the chloroplast trees and the nuclear tree obtained by using wild diploid species and chromosomes 1-10 of the allotetraploid *A. hypogaea*. Certainly, one of them is recombination between homeologous chromosomes, as this event is very common in angiosperm allopolyploids (Deb et al. 2023; Mason and Wendel, 2020). This would have caused the fixation of DNA segments from the B paternal genome in chromosomes 1-10 of cultivated peanut. Importantly, Bertioli et al. (2016) and Bertioli et al (2019) already showed the occurrence of homeologous recombination in *A. hypogaea*, which changed the genomic formula of specific chromosomal regions from the expected AABB to AAAA or BBBB.

Mis-assemblies of homeologous regions in the reference genome can also explain differences between the topologies of Fig. 1b and Supplementary Fig. S4a. In support of this, we point out that we performed SNP calling using the Tifrunner v1 assembly, and the newly released Tifrunner v2 assembly reports several changes in correspondence of homeologous regions (https://www.peanutbase.org/genome/peanut_genome_v1_v2).

We revised the manuscript to make clear all the possible reasons of inconsistencies between the topologies of the chloroplast and nuclear phylogenetic trees, as above mentioned. In addition, we cite the work of Tian et al. (2021), reporting inconsistencies among nuclear peanut phylogenies, and the work of Hodel et al. (2022), highlighting that inconsistencies between nuclear and chloroplast tree topologies have been commonly observed in plants.

2. When aligning WGS to the genome, alignment performance between *Ahh* and *Ahf* needs to be compared to clarify that there are no significant biases.

>>>We acknowledge the reviewer for this useful comment. We calculated the percentage of unique mapped reads associated with *Ahh* and *Ahf* samples and found no significant difference between the two groups (see the new Supplementary Table 8).

3. In addition, the subsequent process of gene screening is still unclear. Following up on the candidate intervals obtained from GWAS and QTL, do the intervals contain only the candidate genes described in the manuscript? The selection criteria of functional genes in the manuscript are still unclear. >>>We understand this point of criticism. In the revised Materials and Methods, we specify that, after QTL mapping, KASP markers were developed within the QTL intervals based on polymorphisms between the parental lines. Then, we used these KASP markers to screen recombinants, allowing further refinement of the QTL interval. In the revised version of the manuscript, we provide new supplementary Figures (S8, S10 and S13) explaining the fine mapping procedure and results. In addition, we provide new Supplementary Tables (Table S13, S20 and S27) indicating the genes residing in the interval identified by fine-mapping, as well as genes for which a mutation could be identified in the cds or in the gene upstream/downstream regions.

As for the flowering pattern, recombinant screening allowed to fine-map the QTL on chr12 in a 514.83 Kb region containing 52 genes. Among them, *AhTFL1* was the only one associated with a frameshift mutation; in addition *AhTFL1* was found to be closely related to *AtTFL1*, involved in the control of inflorescence architecture in *Arabidopsis*; finally, we found full co-segregation between *AhTFL1* mutations and the sequential flowering pattern. Overall, we believe that this body of evidence (and the recent work from Kunta et al. 2022, discussed below) make *AhTFL1* an obvious candidate to control the flowering pattern in peanut.

As for the seed integument colour, fine mapping with RIL recombinant screening resulted in an interval of 540.14 Kb. In the last few months, during the revision of this manuscript, we developed an F₂ population of 7,900 individuals, which was used to further restrict the interval to 107.88 Kb. The interval contains 14 genes. A laccase related to *AtLAC15*, influencing the seed colour in *Arabidopsis*, was the only gene associated with a frameshift mutation, and *AhLAC* mutations were found to co-segregate with the white tegument colour.

Finally, concerning the growth habit, fine mapping with RIL recombinant screening resulted in an interval of 299.11 Kb containing 20 genes. Among them, a *MADS-box* gene was chosen as candidate, as it was the only one displaying a mutation within the cds (a frameshift caused by a 1,870 bp deletion), and the *MADS-box* family of transcription factors was previously associated with the plant growth habit.

4. Regarding the scientific issues about whether the same mother plant contributed to the allopolyploidization of *Ahh* and *Ahf*, can a clear conclusion be drawn based on the existing results?

>>> We think that our results provide solid evidence that *Ahh* and *Ahf* originated from different mother plants belonging to the same species, *A. duranensis*.

Have the authors identified the mother plant?

>>> We are not sure we got the meaning of this question. Clearly, it is not possible to identify today the individual *A. duranensis* plants participating to peanut allopolyploidization. However, based on our findings, we are confident that the plants originating *Ahh* and *Ahf* had a different chloroplast genome.

5. Regarding the claim "wild species within the *Ahh* clade is a remarkable finding indicating that *Ahh* arose from a polyploidization event different from the one originating *Ahf*", the author did not provide a proper response to the reviewer's question. Chloroplast genomes cannot be used to distinguish closely related species and it is not suitable for evolutionary research within species. Using KASP assays to verify the chloroplast markers does not provide an answer to this question.

>>> As stated above, the main purpose of our chloroplast phylogenetic analysis was to investigate inter-specific relationships between the tetraploid *A. hypogaea* and several wild diploid *Arachis* species, thus allowing to reconstruct, from the maternal side, the history of peanut polyploidization, and a body of literature supports our experimental design. We agree with the reviewer that confirming polymorphisms with KASP markers is not a decisive analysis, it was just useful to prove the reliability of the variants we called.

6. A discussion on putative genes (e.g. TFL1) without experimental proof is not convincing. The authors should explore the relevant phenotype (prostrate growth) in other species with a transformation system available.

>>> We realize that functional analyses are necessary to claim that a candidate gene is for sure involved in determining phenotypic variation:

- 1) As for *AhLAC*, we obtained four independent *Arabidopsis* 35S::*AhLAC* transgenic lines, displaying complementation of the *Ahtfl1* mutant phenotype. We understand that developing transgenic lines under the control of the native *AhLAC* promoter would have been a better approach, however developing new transgenic lines would take almost one year and we are afraid that this would significantly affect the novelty of our findings. However, in the revised version of the manuscript, we provided additional evidence that *AhLAC* controls the inner tegument colour, as we correlated the level of the *AhLAC* expression in the transgenic lines with the level of seed darkness, which was measured by a high precision spectrophotometer (see Supplementary Fig. 12).
- 2) As for *AhTFL1*, unfortunately *Arabidopsis* does not show the same flowering pattern of peanut, so complementation by genetic transformation would have not been a suitable approach. However, as we discussed above, *AhTFL1* was the only gene in the QTL confidence interval to be associated with a frameshift mutation, and is related to *AtTFL1*, controlling the flowering pattern in *Arabidopsis*. In addition, we found (and cite in the revised manuscript) the very recent work of Kunta et al. (2022), reporting full co-segregation between the 1492 bp deletion in *AhTFL1* with the alternate flowering pattern, and significant lower expression of *AhTFL1*: a) in *Ahf* compared to *Ahh* and 2) in flowering compared to non-flowering branches. Overall, we believe that our findings, together with those from Kunta et al. (2022), make *AhTFL1* an obvious candidate to control the flowering pattern in peanut. We highlight that, in the revised version of the manuscript, while we write that that *AhTFL1* is an obvious candidate gene for the control of flowering pattern, we avoid to claim that *AhTFL1* is for sure responsible for phenotypic variation.
- 3) As for the *MADS-box* gene associated with the growth habit, complementation in *Arabidopsis* was not performed as *Arabidopsis* does not display the same growth habit phenotypes of peanut. We further modified the manuscript to make clearer that this *MADS-box* gene should be regarded as a candidate. We are available to eliminate the part of the manuscript relative to the growth habit in case the reviewer or the editor would consider it too preliminary to deserve publication in *Nature Genetics*.
6. The data in the revised version still involves the use of 35S constitutive promoter for the incomplete complementation in *Arabidopsis*, and there is no solid evidence to show that *AhLAC* controls the inner tegument color in peanuts. The authors should use a native promoter to generate transgenic lines and perform a biochemical assay to support the changes in the inner tegument color mediated by *AhLAC*.
>>> We understand that developing transgenic lines under the control of the native *AhLAC* promoter would have been a better approach, however developing new transgenic lines would take almost one year and we are afraid that this would significantly affect the novelty of our findings. However, in the revised version of the manuscript, we provided additional evidence that *AhLAC* controls the inner tegument colour, as we correlated the level of the *AhLAC* expression in the transgenic lines with the level of seed darkness, which was measured by a high precision spectrophotometer (see Supplementary Fig. 12c).

Reviewer #2:

Remarks to the Author:

The revised manuscript by Dr. Zheng and colleagues is a vast improvement on the initial submission. The authors clearly took on board my comments to make much more clear the broad impact of their

work, and in this, it is clear that impact goes well outside crop breeders to those working on polyploids generally. They also make clear the interest in the peanut community, addressing an ongoing controversy in the field. Their data and methodology are to my assessment sound and their conclusions are perfectly reasonably worded, representing their findings well. The writing is very clear and engaging. Overall I have no comments for improvement and believe the revised version would be of broad interest.

>>> We take the chance to acknowledge reviewer2 for the time and efforts made in reviewing our manuscript and the nice comments on our work.

Reference

- Tian, X. et al. Chloroplast phylogenomic analyses reveal a maternal hybridization event leading to the formation of cultivated peanuts. *Front. Plant. Sci.* **12**, 804568 (2021).
- Moore, M. J., Bell, C. D., Soltis, P. S. & Soltis, D. E. Using plastid genome-scale data to resolve enigmatic relationships among basal angiosperms. *Proc. Natl. Acad. Sci. U. S. A.* **104**, 19363 (2007).
- Daniell, H., Lin, C. S., Yu, M. & Chang, W.J. Chloroplast genomes: diversity, evolution, and applications in genetic engineering. *Genome Biol.* **17**, 134 (2016).
- Tonti-Filippini, J., Nevill, P. G., Dixon, K. & Small, I. What can we do with 1000 plastid genomes? *Plant J.*, **90**, 808-818 (2017).
- Brock, J. R., Mandáková, T., McKain, M., Lysak, M. A. & Olsen, K. M. Chloroplast phylogenomics in *Camelina* (*Brassicaceae*) reveals multiple origins of polyploid species and the maternal lineage of *C. sativa*. *Hortic Res.* **9**, uhab050 (2022).
- Liu, C. & Musial, J. The application of chloroplast DNA clones in identifying maternal donors for polyploid species of *Stylosanthes*. *Theor. Appl. Genet.* **102**, 73–77 (2001).
- Chen, S. et al. Chloroplast phylogenomic analyses resolve multiple origins of the *Kengyilia* Species (*Poaceae: Triticeae*) via independent polyploidization events. *Front. Plant Sci.* **12**, 682040 (2021).
- Chen, N. et al. Evolutionary patterns of plastome uncover diploid-polyploid maternal relationships in Triticeae. *Mol. Phylo. Evol.* **149**, 106838 (2020).
- Wolfe, T. M. et al. Recurrent allopolyploidizations diversify ecophysiological traits in marsh orchids (*Dactylorhiza majalis* s.l.). *Mol. Ecol.* **32**, 4777-4790 (2023).

- Mavrodiev, E. V. et al. Multiple origins and chromosomal novelty in the allotetraploid *Tragopogon castellanus* (Asteraceae). *New Phytol.* **206**, 1172-1183 (2015).
- Soltis, D. E. & Soltis, P. S. Polyploidy: recurrent formation and genome evolution. *Trends Ecol. Evol.* **14**, 348-352 (1999).
- Mao, L., Zou, Q., Sun, Z., Dong, Q. & Cao, X. Insights into chloroplast genome structure, intraspecific variation, and phylogeny of *Cyclamen* species (*Myrsinoideae*). *Sci. Rep.* **13**, 87 (2023).
- Zhang, W. et al. Comparative analysis of 17 complete chloroplast genomes reveals intraspecific variation and relationships among *Pseudostellaria heterophylla* (Miq.) Pax populations. *Front. Plant Sci.*, **14**, 1163325 (2023).
- Lian, C. et al. Comparative analysis of chloroplast genomes reveals phylogenetic relationships and intraspecific variation in the medicinal plant *Isodon rubescens*. *PLoS ONE.* **17**, e0266546 (2022).
- Mu, Z. et al. Intraspecific chloroplast genome variation and domestication origins of major cultivars of *Styphnolobium japonicum*. *Genes.* **14**, 1156 (2023).
- Deb, S. K., Edger, P. P., Pires, J. C. & McKain, M. R. Patterns, mechanisms, and consequences of homoeologous exchange in allopolyploid angiosperms: a genomic and epigenomic perspective. *New Phytol.* **238**, 2284-2304 (2023).
- Mason, A. S., Wendel, J. F. Homoeologous exchanges, segmental allopolyploidy, and polyploid genome evolution. *Front Genet.* **11**:1014 (2020).
- Bertioli, D. J. et al. The genome sequences of *Arachis duranensis* and *Arachis ipaensis*, the diploid ancestors of cultivated peanut. *Nat. Genet.* **48**, 438-446 (2016).
- Bertioli, D. J. et al. The genome sequence of segmental allotetraploid peanut *Arachis hypogaea*. *Nat. Genet.* **51**, 877-884 (2019).
- Hodel, R. G. J., Zimmer, E. A., Liu, B. B. & Wen, J. Synthesis of nuclear and chloroplast data combined with network analyses supports the polyploid origin of the apple tribe and the hybrid origin of the maleae-gillenieae clade. *Front. Plant Sci.* **12**, 820997 (2022).
- Kunta, S. et al. Identification of a major locus for flowering pattern sheds light on plant architecture diversification in cultivated peanut. *Theor. Appl. Genet.* **135**, 1767-1777 (2022).

Decision Letter, second revision:

18th Oct 2023

Dear Dr Zheng,

Your Article, "DNA sequencing sheds light on peanut evolutionary history and phenotypic diversification" has now been seen by 1 referee. You will see from their comments below that while they find your work of interest, some important points are raised. We are interested in the possibility of publishing your study in Nature Genetics, but would like to consider your response to these concerns in the form of a revised manuscript before we make a final decision on publication.

We therefore invite you to revise your manuscript taking into account all reviewer and editor comments. Please highlight all changes in the manuscript text file. At this stage we will need you to upload a copy of the manuscript in MS Word .docx or similar editable format.

*2) If you have not done so already please begin to revise your manuscript so that it conforms to our Article format instructions, available here.

*3) Include a revised version of any required Reporting Summary:

Please be aware of our guidelines on digital image standards.

Note: This URL links to your confidential home page and associated information about manuscripts you may have submitted, or that you are reviewing for us. If you wish to forward this email to co-

authors, please delete the link to your homepage.

We hope to receive your revised manuscript within four to eight weeks. If you cannot send it within this time, please let us know.

Sincerely,
Wei

Wei Li, PhD
Senior Editor
Nature Genetics
New York, NY 10004, USA
www.nature.com/ng

Reviewers' Comments:

Reviewer #1:

Remarks to the Author:

The manuscript has improved but some issues remain:

In the revised version of the manuscript, the authors extended the analysis of chloroplast variation using MNR markers to support the tree using KASP markers (Fig1b). In Lines 87-88, the 113 assembled chloroplast genomes were processed to identify MNR loci. However, in Figure S3, only 93 accessions were used; why the sample number is inconsistent with that in Figure 1b?

While there is a lack of reports on the isoflavone compounds contributing to the inner tegument color in peanuts, the functional complementation in Arabidopsis cannot infer that AhLAC is causing the pigmentation through "accumulation of oxidized polymeric forms of flavonoids" in peanuts (Line 285-286), unless the roles of such forms of flavonoids are verified in peanut. Again, the partial

complementation may suggest the role of AhLAC in peanuts differs from that in the ectopic system. Moreover, the author should also analyze the expression level of AhLAC in peanuts.

Author Rebuttal, second revision:

Reviewer #1:

Remarks to the Author:

The manuscript has improved but some issues remain:

In the revised version of the manuscript, the authors extended the analysis of chloroplast variation using MNR markers to support the tree using KASP markers (Fig1b). In Lines 87-88, the 113 assembled chloroplast genomes were processed to identify MNR loci. However, in Figure S3, only 93 accessions were used; why the sample number is inconsistent with that in Figure 1b?

>>> We are glad to know that our revisions were appreciated. Indeed, it was an oversight on our part, as the figure was omitting distantly related wild species. We acknowledge the reviewer for this remark. In the revised version of the manuscript, Figure S3 was modified to include all the 113 chloroplast genomes. As expected, genetic relationship within *A.hypogaea*, and between *A. hypogaea* and closely related wild species are the same as those shown in the previous version of the figure.

While there is a lack of reports on the isoflavone compounds contributing to the inner tegument color in peanuts, the functional complementation in *Arabidopsis* cannot infer that AhLAC is causing the pigmentation through "accumulation of oxidized polymeric forms of flavonoids" in peanuts (Line 285-286), unless the roles of such forms of flavonoids are verified in peanut. Again, the partial complementation may suggest the role of AhLAC in peanuts differs from that in the ectopic system. Moreover, the author should also analyze the expression level of AhLAC in peanuts.

>>>We acknowledge the reviewer for this suggestion. We quantified the expression of *AhLAC* in the tegument tissues of four peanut genotypes, two displaying the MITE insertion in the *cds* and the white tegument color, and two displaying no MITE insertion and yellow tegument color. As shown in the new Fig.3i, *AhLAC* expression was markedly higher in yellow tegument genotypes, consistent with a role of *AhLAC* in determining tegument pigmentation.

Mutation in *Arabidopsis AtTT10* was shown to be associated with the oxidative polymerization of epicatechin in the yellow dimer dehydrodiepicatechin A (Pourcel et al. 2009 Plant Cell 17:2966-2980). Therefore, we quantified epicatechin in the seed coat of genotypes with different tegument color. As shown in the new Supplementary Figures 12d-e, we detected significantly higher levels of epicatechins in genotypes displaying white tegument, which is consistent with a role of AhLAC in causing pigmentation through epicatechin oxidative polymerization, similarly to *AtTT10*. However, we agree with the reviewer that the enzymatic activity of *AhLAC* should be further investigated, therefore we modified the discussion with the following statement: "further investigation is needed to clarify whether *AhLAC* promotes tegument pigmentation through the oxidative polymerization of flavonoids, as it was shown for its *Arabidopsis* homolog *Attt10* (Pourcel et al. 2005)".

Decision Letter, third revision:

8th Apr 2024

Dear Dr. Zheng,

Thank you for submitting your revised manuscript "DNA sequencing sheds light on peanut evolutionary history and phenotypic diversification" (NG-A60406R2). It has now been seen by the original referees and their comments are below. The reviewers find that the paper has improved in revision, and therefore we'll be happy in principle to publish it in Nature Genetics, pending minor revisions to comply with our editorial and formatting guidelines.

Sincerely,
Wei

Wei Li, PhD
Senior Editor
Nature Genetics
New York, NY 10004, USA
www.nature.com/ng

Reviewer #1 (Remarks to the Author):

The authors have addressed all my questions and comments.

Final Decision Letter:

18th Jul 2024

Dear Dr. Zheng,

I am delighted to say that your manuscript "Chloroplast and whole-genome sequencing sheds light on peanut evolutionary history and phenotypic diversification" has been accepted for publication in an upcoming issue of Nature Genetics.

Over the next few weeks, your paper will be copyedited to ensure that it conforms to Nature Genetics

style. Once your paper is typeset, you will receive an email with a link to choose the appropriate publishing options for your paper and our Author Services team will be in touch regarding any additional information that may be required.

Your paper will be published online after we receive your corrections and will appear in print in the next available issue. You can find out your date of online publication by contacting the Nature Press Office (press@nature.com) after sending your e-proof corrections.

Please note that *Nature Genetics* is a Transformative Journal (TJ). Authors may publish their research with us through the traditional subscription access route or make their paper immediately open access through payment of an article-processing charge (APC). Authors will not be required to make a final decision about access to their article until it has been accepted. Find out more about Transformative Journals

Authors may need to take specific actions to achieve compliance with funder and institutional open access mandates. If your research is supported by a funder that requires immediate open access (e.g. according to Plan S principles) then you should select the gold OA route, and we will direct you to the compliant route where possible. For authors selecting the subscription

publication route, the journal's standard licensing terms will need to be accepted, including <https://www.nature.com/nature-portfolio/editorial-policies/self-archiving-and-license-to-publish>. Those licensing terms will supersede any other terms that the author or any third party may assert apply to any version of the manuscript.

If you have not already done so, we strongly recommend that you upload the step-by-step protocols used in this manuscript to protocols.io. protocols.io is an open online resource that allows researchers to share their detailed experimental know-how. All uploaded protocols are made freely available and are assigned DOIs for ease of citation. Protocols can be linked to any publications in which they are used and will be linked to from your article. You can also establish a dedicated workspace to collect all your lab Protocols. By uploading your Protocols to protocols.io, you are enabling researchers to more readily reproduce or adapt the methodology you use, as well as increasing the visibility of your protocols and papers. Upload your Protocols at <https://protocols.io>. Further information can be found at <https://www.protocols.io/help/publish-articles>.

Sincerely,
Wei

Wei Li, PhD
Senior Editor
Nature Genetics
www.nature.com/ng